# ReCalKV: Low-Rank KV Cache Compression via Head Reordering and Offline Calibration

## Abstract

Large language models (LLMs) have demonstrated remarkable performance, but their long-context reasoning remains constrained by the excessive memory required for the Key-Value (KV) cache. This makes KV cache compression a critical step toward efficient long-context inference. Recent methods have explored low-rank techniques to reduce the hidden size of the KV cache. However, they neglect the distinct roles and varying importance of Keys and Values, leading to significant performance drops under high compression. To address this, we propose ReCalKV, a post-training low-rank KV cache compression approach with tailored strategies for Keys and Values. For Keys, we propose Head-wise Similarity–aware Reordering (HSR), which clusters structurally similar heads into groups, enabling more accurate low-rank approximation via grouped SVD. For Values, we propose Offline Value Calibration (OVC), which efficiently calibrates the value projection matrix using calibration data without training, ensuring an accurate representation of contextual information. Extensive experiments show that ReCalKV consistently outperforms existing low-rank compression methods, achieving high compression ratios with minimal performance loss. We will release all the code and models.

## 1 Introduction

Large language models (LLMs) (Vaswani, 2017; Touvron et al., 2023a; Dubey et al., 2024) have demonstrated outstanding performance across a wide range of tasks. To accelerate inference, modern LLMs cache intermediate Key-Value (KV) states, avoiding redundant computation during autoregressive decoding. However, as the input context length increases, the KV cache grows rapidly, leading to substantial memory overhead and bandwidth pressure. In practice, the KV cache often becomes the primary bottleneck for long-context inference. Consequently, compressing the KV cache becomes essential for enabling efficient and scalable deployment of LLMs across real-world applications.

To reduce the size of the KV cache, recent works explore compression along multiple axes. Multi-query attention (Shazeer, 2019) and grouped-query attention (Ainslie et al., 2023) reduce the number of heads by sharing keys and values. Quantization methods (Zhao et al., 2023) lower KV cache's precision, with some (Liu et al., 2024c; Hooper et al., 2024) pushing KV representations down to 2 bits. Others (Zhang et al., 2023; Li et al., 2024; Xiao et al., 2023; Dong et al., 2024) reduce the number of cached tokens by selecting only important ones, often based on attention scores. A few methods (Chang et al., 2025; Liu et al., 2024b) further compress across layers by reusing KV states. These methods reveal the multi-dimensional structure of KV cache compression.

Another line of work (Chang et al., 2024; Liu et al., 2024a; Lin et al., 2024) explores KV cache compression from a different angle—by reducing the dimensionality of the hidden vector space of Keys and Values themselves. For example, MLA (Liu et al., 2024a) reduces memory via low-rank representations but requires training the model from scratch. Other approaches, such as EigenAttention (Saxena et al., 2024) and MastryohakaKV (Lin et al., 2024), compress KV entries via projection into lower-dimensional subspaces. Although effective, their projection and reconstruction introduce extra decoding overhead, limiting applicability in latency-sensitive scenarios. Palu (Chang et al., 2024) and LoRC (Zhang et al., 2024) address this issue by directly applying Singular Value Decomposition (SVD) to the KV projection layers, effectively compressing the hidden dimensions of the KV cache. This approach substantially reduces the overhead of runtime projection and reconstruction. However, both methods overlook the inherent asymmetry between Keys and Values in the attention mechanism, and their performance degrades notably under high KV cache compression ratios.

To better explore KV cache compression along the hidden dimension, we conduct detailed analyses of the roles of Keys and Values in the attention mechanism. Our analyses reveal that: **(i)** Most modern LLMs (Touvron et al., 2023a;b; Dubey et al., 2024) use positional encoding, typically RoPE (Su et al., 2024), which is applied to Keys. As a result, low-rank compressed Keys must be fully reconstructed during inference to enable positional encoding, which introduces additional computational overhead. This makes it essential to consider both accuracy and computational cost when compressing Key cache. **(ii)** We measure the Fisher information of the Key and Value projection layers. Our Fisher information analysis reveals that the Value projection matrices carry significantly higher importance than their Key counterparts, highlighting the crucial role of Value representations in the overall model behavior. Therefore, minimizing accuracy degradation is critical when compressing the Value cache.

Based on our analyses, we develop distinct compression strategies for Keys and Values based on their different roles and varying importance in the attention mechanism. For Keys, we propose **Head-wise Similarity–aware Reordering** (HSR). It first reorders attention heads based on their representation similarity, then groups similar heads together, and applies grouped SVD within each group. This reduces the Key cache size with low reconstruction overhead. Grouping similar heads helps the SVD better capture shared subspace structures, lowering approximation error and preserving accuracy. For Values, we propose **Offline Value Calibration** (OVC). We first apply SVD to the Value projection matrix and then calibrate the decomposed components with a small calibration dataset to preserve Value accuracy. We also fuse the right factor of the SVD decomposition into the subsequent output projection matrix, removing the need for explicit reconstruction during inference.

Extensive experiments demonstrate that ReCalKV consistently achieves SOTA performance across multiple LLM families, clearly surpassing existing low-rank compression methods under various evaluation settings. For example, on the LLaMA-2-7B model (Touvron et al., 2023b) evaluated on six zero-shot QA datasets, ReCalKV achieves an average accuracy of 63.64% under a 50% KV cache compression ratio, compared to 64.99% for the full-precision model—corresponding to only a 2% relative drop. Notably, since ReCalKV is orthogonal to quantization techniques, it can be seamlessly integrated with them to achieve even higher overall compression ratios.

Our key contributions can be summarized as follows:

- We propose ReCalKV, a novel post-training KV cache compression framework that reduces memory overhead in long-context inference without requiring model retraining.

- We propose the Head-wise Similarity–aware Reordering (HSR) strategy for Key compression, which effectively reduces the Key cache size with limited reconstruction overhead.

- We propose an Offline Value Calibration (OVC) strategy for compressing the Value cache, preserving accuracy without introducing additional inference overhead.

- Extensive experiments demonstrate that ReCalKV consistently outperforms prior low-rank compression approaches. Furthermore, ReCalKV can be seamlessly combined with quantization techniques to achieve even higher compression ratios.

## 2 RELATED WORK

**SVD-Based LLM Compression.** Singular Value Decomposition (SVD) has been widely adopted for compressing LLMs by approximating weight matrices with low-rank factors. However, standard SVD (Noach & Goldberg, 2020) has been observed to cause notable performance degradation in practice. To address this, FWSVD (Hsu et al., 2022) incorporates Fisher information to guide decomposition, while ASVD (Yuan et al., 2023) rescales weights to mitigate the impact of activation outliers. SVD-LLM (Wang et al., 2024) establishes a direct connection between singular values and compression loss, employing data whitening and selective truncation to minimize degradation. AdaSVD (Li et al., 2025) further improves compression by introducing adaptive rank allocation and compensation mechanisms. Unlike prior methods that target model weights, we apply SVD to the key and value projections to compress the KV cache directly.

**KV Cache Compression.** To support long-context inference, various methods have been proposed to compress the KV cache. Quantization-based approaches are widely adopted: Atom (Zhao et al., 2023) performs per-token quantization, while WKVQuant (Yue et al., 2024) uses a two-level scheme for better accuracy. KIVILiu et al. (2024c) and KVQuant (Hooper et al., 2024) combine per-token and per-channel quantization, with KVQuant further leveraging non-uniform and sparse techniques

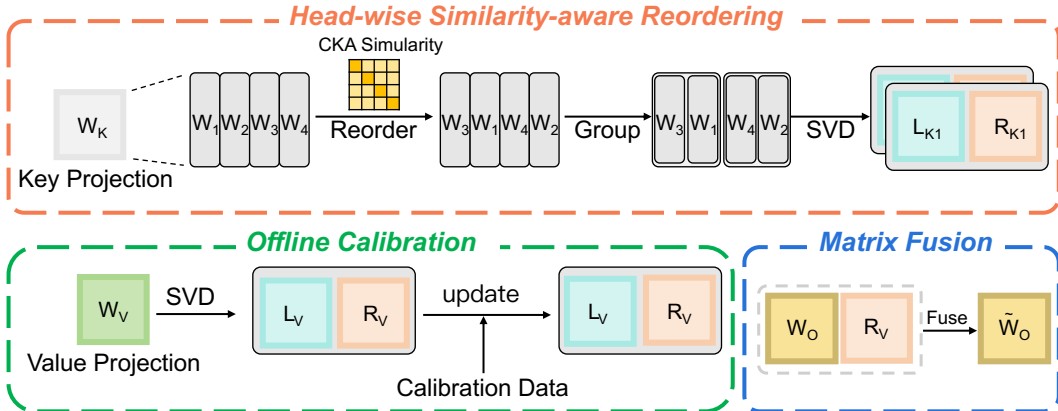

Figure 1: **Overview of the ReCalKV framework.** The method consists of three key components: Head-wise Similarity-aware Reordering (HSR) for compressing Keys via grouped SVD, and Offline Value Calibration (OVC) for compressing Values without additional runtime overhead.

to handle outliers. In parallel, token eviction methods (Zhang et al., 2023; Li et al., 2024; Xiao et al., 2023; Dong et al., 2024) reduce memory by discarding less relevant tokens or retrieving only subsets during decoding. Beyond these strategies, several methods aim to reduce the hidden dimension of KV representations. DeepSeek V2 (Liu et al., 2024a) uses MLA for built-in dimension reduction, but requires training from scratch. MatryoshkaKV (Lin et al., 2024) and Eigen-Attention (Saxena et al., 2024) project the KV cache into a low-rank space via additional projection matrices, at the cost of increased computation. HeadKV (Fu et al., 2024) uses SVD to reduce the number of KV heads, leading to improved efficiency. LoRC (Zhang et al., 2024) and Palu (Chang et al., 2024) directly apply SVD to the KV projection matrices, reducing dimensionality with minimal architectural changes. Our method is a post-training low-rank KV compression approach and is orthogonal to quantization and token eviction, enabling easy integration for further compression gains.

## 3 METHODOLOGY

### 3.1 PRELIMINARY

**Singular Value Decomposition.**

Singular Value Decomposition (Golub et al., 1987) is a classical method for obtaining low-rank approximations. Given $\mathbf{W} \in \mathbb{R}^{m \times n}$, it factorizes the matrix as $\mathbf{W} = \mathbf{U}\boldsymbol{\Sigma}\mathbf{V}^\top$, where $\mathbf{U}$ and $\mathbf{V}$ contain the left and right singular vectors, and $\boldsymbol{\Sigma}$ holds the singular values. Keeping only the top-$r$ singular values and vectors yields the low-rank approximation:

$$\mathbf{W} \approx \mathbf{LR}, \quad \text{where} \quad \mathbf{L} = \mathbf{U}_r\boldsymbol{\Sigma}_r^{1/2}, \quad \mathbf{R} = \boldsymbol{\Sigma}_r^{1/2}\mathbf{V}_r^\top. \tag{1}$$

Here, $\mathbf{U}_r \in \mathbb{R}^{m \times r}$ and $\mathbf{V}_r \in \mathbb{R}^{n \times r}$ are the top-$r$ singular vectors of $\mathbf{W}$, and $\boldsymbol{\Sigma}_r \in \mathbb{R}^{r \times r}$ contains the corresponding singular values. This yields two smaller matrices, $\mathbf{L}$ and $\mathbf{R}$, from the low-rank decomposition of $\mathbf{W}$.

Given an input $\mathbf{x} \in \mathbb{R}^{1 \times m}$, we compute the intermediate representation $\mathbf{z} = \mathbf{xL}$ and store $\mathbf{z} \in \mathbb{R}^{1 \times r}$ in the KV cache instead of the full output. During attention, the output is approximated by reconstructing $\mathbf{xW} \approx \mathbf{zR}$. This approach reduces the KV cache size with a compression ratio of $r/n$, while maintaining a close approximation of the original computation.

To further enhance decomposition stability, we follow SVD-LLM (Wang et al., 2024) and apply a whitening step before performing SVD. Specifically, we compute the input and output activation covariance matrices from calibration data and construct whitening matrices that normalize their statistical distributions. The weight matrix is first transformed into a whitened space with more isotropic activations, SVD is then applied to the whitened weight, and the resulting factors are mapped back to the original space. This procedure reduces condition numbers, mitigates sensitivity to outliers, and yields more accurate low-rank reconstructions.

**Centered Kernel Alignment Similarity.** Centered Kernel Alignment (CKA) (Kornblith et al., 2019) is a widely used metric for quantifying the similarity between two sets of representations. Given two matrices $\mathbf{X} \in \mathbb{R}^{n \times d_1}$ and $\mathbf{Y} \in \mathbb{R}^{n \times d_2}$, CKA is computed by first forming their Gram (kernel)

matrices $\mathbf{G_X} = \mathbf{XX}^\top$ and $\mathbf{G_Y} = \mathbf{YY}^\top$. These kernel matrices are then centered as follows:

$$\tilde{\mathbf{G}}_\mathbf{X} = \mathbf{HG_XH}, \quad \tilde{\mathbf{G}}_\mathbf{Y} = \mathbf{HG_YH}, \tag{2}$$

where $\mathbf{H} = \mathbf{I}_n - \frac{1}{n}\mathbf{1}_n\mathbf{1}_n^\top$ is the centering matrix. The final CKA similarity is defined as:

$$\text{CKA}(\mathbf{X}, \mathbf{Y}) = \frac{\text{HSIC}(\mathbf{X}, \mathbf{Y})}{\sqrt{\text{HSIC}(\mathbf{X}, \mathbf{X}) \cdot \text{HSIC}(\mathbf{Y}, \mathbf{Y})}}, \tag{3}$$

where $\text{HSIC}(\mathbf{X}, \mathbf{Y}) = \text{Tr}(\tilde{\mathbf{G}}_\mathbf{X}\tilde{\mathbf{G}}_\mathbf{Y})$ denotes the Hilbert-Schmidt Independence Criterion (HSIC). CKA ranges from 0 to 1, with higher values indicating greater similarity between the two matrices.

## 3.2 HEAD-WISE SIMILARITY-AWARE REORDERING

**Group-head Low-rank Decomposition.** Following the grouped decomposition strategy proposed in Palu (Chang et al., 2024), we organize multiple attention heads into groups prior to performing SVD. Given a Key projection matrix $\mathbf{W} \in \mathbb{R}^{m \times n}$, where $n = h \cdot d_h$ corresponds to $h$ attention heads each of hidden dimension $d_h$, we divide $\mathbf{W}$ column-wise into $h$ submatrices, each representing a single head. Then we group every $s$ heads into one group, result-

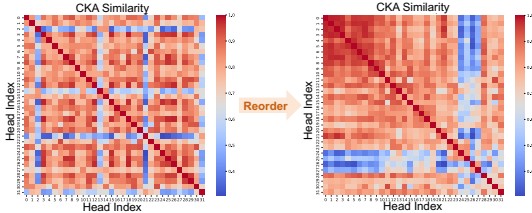

Figure 2: CKA similarity matrices before and after head reordering.

ing in $g = h/s$ groups in total. For each group $j$, we construct a concatenated projection matrix $\mathbf{W}_{g_j} = \left[\mathbf{W}_{j,1}, \ldots, \mathbf{W}_{j,s}\right]$, where each $\mathbf{W}_{j,k} \in \mathbb{R}^{m \times d_h}$ is the projection matrix of the $k$-th head in group $j$, and thus $\mathbf{W}_{g_j} \in \mathbb{R}^{m \times (s \cdot d_h)}$. Instead of applying SVD to the entire projection matrix at once, we apply low-rank approximation to the grouped matrix: $\mathbf{W}_{g_j} \approx \mathbf{L}_{g_j}\mathbf{R}_{g_j}$, where $\mathbf{L}_{g_j} \in \mathbb{R}^{d \times r_g}$ and $\mathbf{R}_{g_j} \in \mathbb{R}^{r_g \times (d_h \cdot s)}$. During inference, the latent representation shared across all heads in the group is computed as: $\mathbf{z}_{g_j} = \mathbf{x}\mathbf{L}_{g_j}$, and the projected outputs for individual heads are reconstructed via:

$$[\mathbf{y}_{j,1}, \ldots, \mathbf{y}_{j,s}] = \mathbf{z}_{g_j}\mathbf{R}_{g_j}. \tag{4}$$

This grouped strategy provides a good trade-off between reconstruction overhead and approximation fidelity, enabling efficient compression with minimal performance impact.

**CKA-based Head Similarity.** A key question in grouped SVD is how to assign attention heads into groups so as to minimize the reconstruction error. Empirically, we observe that grouping heads with similar left singular subspaces results in lower approximation error, as these heads tend to share more common representational components and thus benefit from joint compression. To quantify head similarity, we adopt centered kernel alignment (CKA) (Kornblith et al., 2019), a robust and widely used

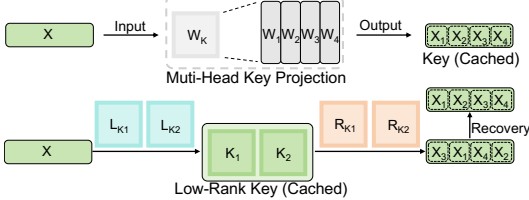

Figure 3: **Key decoding with HSR.** Similar heads are reordered and grouped before SVD, enabling more accurate reconstruction.

metric for comparing representation subspaces. We compute the pairwise CKA similarity between all attention heads, yielding a symmetric similarity matrix $\mathbf{S} \in \mathbb{R}^{h \times h}$, where $h$ is the number of heads:

$$\mathbf{S}_{i,j} = \text{CKA}(\mathbf{H}_i, \mathbf{H}_j), \quad \forall i, j \in 1, \ldots, h. \tag{5}$$

**Head Reordering.** Based on the similarity matrix $\mathbf{S}$, we perform head grouping by prioritizing pairs with high mutual similarity. Specifically, we adopt a greedy strategy that iteratively selects the head pair with the highest CKA similarity and assigns them to the same group, subject to a fixed group size constraint (e.g., 4 heads per group when $h = 32$). Remaining unassigned heads are then added to existing groups with available capacity, ensuring that all heads are eventually grouped. This head reordering process encourages heads with similar representational structure to share SVD decompositions, effectively reducing approximation error during grouped compression. As shown in Figure 2, we visualize the CKA similarity matrices before and after head reordering. It can be observed that, after reordering, heads assigned to adjacent positions exhibit higher mutual similarity, indicating that similar heads are more effectively grouped together. By aligning structurally coherent heads within each group, we improve both the compactness and accuracy of the resulting low-rank representation. To ensure decoding equivalence, we apply an inverse reordering to restore the original head order (see Figure 3); related theoretical analysis is given in the supplementary file.

### 3.3 OFFLINE VALUE CALIBRATION

**Offline Calibration.** For Value cache compression, we directly perform SVD on the full Value projection matrix $\mathbf{W}_v \in \mathbb{R}^{m \times n}$, resulting in the decomposed matrices: $\mathbf{W}_v \approx \mathbf{L}_v \mathbf{R}_v$, where $\mathbf{L}_v \in \mathbb{R}^{m \times r}$ and $\mathbf{R}_v \in \mathbb{R}^{r \times n}$ represent the compressed components of the original Value projection matrix $\mathbf{W}_v \in \mathbb{R}^{m \times n}$. We then define the approximation error $\mathcal{E}$ introduced by the SVD decomposition as:

$$\mathcal{E} = ||\mathbf{L}_v \mathbf{R}_v \mathbf{X} - \mathbf{W}_v \mathbf{X}||_F^2, \tag{6}$$

where $\mathbf{X}$ denotes the calibration dataset. Based on our analysis of Fisher Information, we observe that the Value projection matrix exhibits significantly higher Fisher Information compared to the Key projection matrix. This indicates that the Value projection matrix plays a more critical role in model performance. Therefore, we aim to minimize the approximation error $\mathcal{E}$ introduced during the compression of the Value projection to preserve model accuracy as much as possible. Inspired by Li et al. (2025), we observe that standard SVD decomposition does not always yield the lowest approximation error. To this end, we perform offline calibration of the decomposed matrices $\mathbf{L}_v$ and $\mathbf{R}_v$ using a small calibration dataset $\mathbf{X}$, aiming to further reduce the compression-induced approximation error. We first calibrate $\mathbf{L}_v$ by setting the derivative of the approximation error $\mathcal{E}$ with respect to $\mathbf{L}_v$ to zero:

$$\frac{\partial \mathcal{E}}{\partial \mathbf{L}_v} = 0 \quad \Rightarrow \mathbf{L}_v = \mathbf{W} \mathbf{X} \mathbf{X}^\top \mathbf{R}_v^\top (\mathbf{R}_v \mathbf{X} \mathbf{X}^\top \mathbf{R}_v)^{-1}. \tag{7}$$

Next, to improve the fidelity of the low-rank approximation, we calibrate the right factor $\mathbf{R}_v$ by minimizing the approximation error. Specifically, we compute the gradient of the approximation error $\mathcal{E}$ with respect to $\mathbf{R}_v$ and set it to zero, allowing us to solve for a more accurate $\mathbf{R}_v$ in closed form:

$$\frac{\partial \mathcal{E}}{\partial \mathbf{R}_v} = 0 \quad \Rightarrow \mathbf{R}_v = ((\mathbf{L}_v)^\top \mathbf{L}_v)^{-1} (\mathbf{L}_v)^\top \mathbf{W}. \tag{8}$$

This yields an optimal closed-form solution for $\mathbf{R}_v$. Together, the calibrated $\mathbf{L}_v$ and $\mathbf{R}_v$ form a refined low-rank approximation of the original Value projection matrix, effectively reducing compression-induced error without incurring any additional inference-time computation. The derivation, closed-form updates for $\mathbf{L}_v$ and $\mathbf{R}_v$, implementation details, related theoretical analysis, and the impact of different calibration datasets and dataset sizes are discussed in the supplementary material.

---

**Algorithm 1** Pseudocode of ReCalKV

---

1: **Inputs:** Model $\mathcal{M}$, Calibration Data $\mathcal{X}$, Target Compression Ratio $\mathcal{TR}$
2: **Output:** $\mathcal{M}'$: Model equipped with compressed KV cache
3: **procedure** RECALKV($\mathcal{M}, \mathcal{X}, \mathcal{TR}$)
4:     $\mathcal{F} \leftarrow$ CALCULATE_FISHER_INFO($\mathcal{M}, \mathcal{X}$)
5:     $R \leftarrow$ ALLOCATE_COMPRESSION_RATIO($\mathcal{M}, \mathcal{TR}, \mathcal{F}$)
6:     **for** each Key projection layer $\mathcal{W}_k$ in model $\mathcal{M}$ **do**
7:         $\mathcal{CKA} \leftarrow$ CALCULATE_CKA_SIMILARITY($\mathcal{W}_k$)
8:         $\mathcal{W}_k' \leftarrow$ HEAD_REORDER($\mathcal{W}_k, \mathcal{CKA}$)
9:         $\mathbf{L}_k[], \mathbf{R}_k[] \leftarrow$ GROUP_SVD($\mathcal{W}_k', \mathcal{R}[\mathcal{W}_k]$)
10:        $\mathcal{M}' \leftarrow$ UPDATE_LAYER($\mathbf{L}_k[] \& \mathbf{R}_k[], \mathcal{W}_k, \mathcal{M}$)
11:     **end for**
12:     **for** each Value projection layer $\mathcal{W}_v$ in model $\mathcal{M}$ **do**
13:         $\mathbf{L}_v, \mathbf{R}_v \leftarrow$ SVD($\mathcal{W}_v, \mathcal{R}[\mathcal{W}_v]$)
14:         $\mathbf{L}_v', \mathbf{R}_v' \leftarrow$ OFFLINE_CALIBRATION($\mathcal{W}_v, \mathbf{L}_v, \mathbf{R}_v, \mathcal{X}$)
15:        $\widetilde{\mathcal{W}_o} \leftarrow$ MATRIX_FUSION($\mathbf{R}_v', \mathcal{W}_o$)
16:        $\mathcal{M}' \leftarrow$ UPDATE_LAYER($\mathbf{L}_v', \mathcal{W}_v, \mathcal{M}$)
17:        $\mathcal{M}' \leftarrow$ UPDATE_LAYER($\mathcal{W}_o', \mathcal{W}_o, \mathcal{M}$)
18:     **end for**
19:     **return** $\mathcal{M}'$
20: **end procedure**

---

**Matrix Fusion.** After performing SVD and offline calibration on the Value projection matrix, we further optimize the inference efficiency by eliminating unnecessary reconstruction steps. Specifically, instead of computing the full Value cache and then applying the output projection matrix $W_o$, we fuse $\mathbf{R}_v$ into $W_o$ to avoid runtime reconstruction. We start from the standard attention output:

$$\text{Output} = \text{Attention}(Q, K, V) W_o = \text{Attention}(X W_q, X W_k, X W_v) W_o. \tag{9}$$

where $X$ is the input sequence, $W_q$, $W_k$, and $W_v$ are the projection matrices for query, key, and value respectively, and $W_o$ is the output projection matrix. For compressed Value cache, we store $X' = X\mathbf{L}_v$ as the low-rank representation. With Value compression, the attention output becomes:

$$\text{Output} = \text{Attention}(XW_q, XW_k, X\mathbf{L}_v\mathbf{R}_v)W_o. \tag{10}$$

We then define a fused output projection matrix $\widetilde{W}_o = \mathbf{R}_v W_o$, and compute the attention output as:

$$\text{Output} = \text{Attention}(XW_q, XW_k, X\mathbf{L}_v)\widetilde{W}_o. \tag{11}$$

This design eliminates the need to explicitly reconstruct $X'\mathbf{R}_v$ during inference. By merging the computation into a single step, matrix fusion reduces both memory usage and computational overhead, streamlining the runtime execution path. Moreover, since the fused matrix $\widetilde{W}_o$ can be precomputed entirely offline, it introduces no additional overhead to online inference, making this approach highly suitable for latency-sensitive applications and deployment on resource-constrained hardware.

### 3.4 ReCalKV Workflow

The overall pipeline of ReCalKV is outlined in Algorithm 1. Given a pre-trained model $\mathcal{M}$, calibration data $\mathcal{X}$, and a target compression ratio $\mathcal{R}$, ReCalKV applies differentiated compression strategies to Key and Value projection layers. We first compute layer-wise Fisher Information scores using the calibration data, following the strategy introduced in Palu (Chang et al., 2024), to estimate the relative importance of each layer and guide compression ratio allocation. Based on this, we allocate compression ratios to different layers accordingly. For Key projection layers, we first compute head-wise CKA similarity, then reorder the heads to group the most similar ones together, and finally apply grouped SVD for compression. For Value projection layers, we apply standard SVD followed by offline calibration to adjust the low-rank factors. Then we fuse the right factor of the decomposition with the output projection matrix $W_o$ in the attention block. The modified matrices are then written back into the model. The final output is a modified model $\mathcal{M}'$ that generates compressed KV caches.

## 4 Experiments

### 4.1 Experimental Settings

**Models.** We evaluate our method on a range of widely adopted LLMs, including multiple generations of the LLaMA family: LLaMA-7B (Touvron et al., 2023a), LLaMA-2-7B (Touvron et al., 2023b), and LLaMA-2-13B-Chat (Touvron et al., 2023b). We also include instruction-tuned variants such as Mistral-7B-Instruct-v0.2 (Jiang et al., 2023) and LongChat-7B-v1.5-32k (Li et al., 2023) to assess performance under extended context settings. These models cover both base and chat/instruction-following variants, ensuring a comprehensive evaluation across different architectures and use cases. Notably, Mistral-7B-Instruct-v0.2 (Jiang et al., 2023) adopt Grouped Query Attention (GQA) (Ainslie et al., 2023), while the others use standard Multi-Head Attention (MHA) (Vaswani, 2017). Additional results on the more recent LLaMA-3.1 model are provided in the supplementary file.

**Datasets and Evaluation.** We assess the effectiveness of our method using both perplexity and task-specific accuracy. For language modeling evaluation, we report perplexity on WikiText2 (Merity et al., 2017), Penn Treebank (PTB)(Plotz & Roth, 2017), and a subset of the C4 corpus(Raffel et al., 2020b). To evaluate reasoning and generalization capabilities, we measure zero-shot accuracy on six QA benchmarks: ARC-c, ARC-e (Clark et al., 2018b), Hellaswag (Zellers et al., 2019b), OBQA (Mihaylov et al., 2018), PIQA (Bisk et al., 2020), and Winogrande (Sakaguchi et al., 2020). In addition, we adopt the LongBench benchmark (Bai et al., 2023) to evaluate long-context performance, conducting experiments on eight diverse tasks that thoroughly evaluate long-context capability.

**Baselines.** We compare our method with Palu (Chang et al., 2024), a recent low-rank compression approach for KV cache. Specifically, we adopt its G-LRD variant for evaluation. For fair comparison, we adopt the same group-wise decomposition with a fixed group size of 4. Further comparisons with ASVD (Yuan et al., 2023) and EigenAttention (Saxena et al., 2024) are included in the supplementary.

**Implementation Details.** All experiments are conducted using PyTorch (Paszke et al., 2019b) and Huggingface Transformers (Paszke et al., 2019a) on a single NVIDIA A800 GPU with 80GB of memory. Following the setup in SVD-LLM (Wang et al., 2024), we apply a whitening transformation before performing SVD truncation. Specifically, we randomly select 256 samples from the WikiText-2 dataset as calibration data and use them both for whitening in the SVD step and for the offline calibration process in Value compression.

Table 1: Zero-shot performance comparison between **ReCalKV** and Palu (Chang et al., 2024) under 50% to 70% compression ratios. Evaluation on three language modeling datasets (measured by perplexity (↓)) and six zero-shot QA datasets (measured by both individual and average accuracy (↑)).

| Ratio | Method | Wiki2↓ | PTB↓ | C4↓ | OBQA | Hella | PIQA | ARC-e | ARC-c | Wino | Average↑ |
|---|---|---|---|---|---|---|---|---|---|---|---|
| | | | | | **LLaMA-7B** (Touvron et al., 2023a) | | | | | | |
| 0% | Original | 5.68 | 41.15 | 7.34 | 44.40 | 76.18 | 78.67 | 75.25 | 44.80 | 70.01 | 64.89 |
| 50% | Palu | 6.27 | 48.39 | 8.85 | 41.6 | 73.46 | 76.71 | 72.35 | 40.53 | 68.75 | 62.23 |
| | **ReCalKV** | **6.13** | **43.99** | **8.36** | **42.00** | **73.59** | **77.20** | **72.35** | **41.72** | **68.35** | **62.54** |
| 60% | Palu | 7.08 | 91.45 | 10.98 | 36.80 | 68.80 | 73.94 | 67.63 | 38.14 | 63.61 | 58.15 |
| | **ReCalKV** | **6.63** | **56.49** | **9.44** | **39.80** | **71.51** | **75.95** | **71.09** | **40.10** | **64.17** | **60.44** |
| 70% | Palu | 8.42 | 211.33 | 14.46 | 34.60 | 61.06 | 71.00 | 61.53 | 33.70 | 59.51 | 53.57 |
| | **ReCalKV** | **7.24** | **71.47** | **10.53** | **38.20** | **68.73** | **75.19** | **67.55** | **39.08** | **64.01** | **58.79** |
| | | | | | **LLaMA-2-7B** (Touvron et al., 2023b) | | | | | | |
| 0% | Original | 5.47 | 37.91 | 7.26 | 44.20 | 76.01 | 78.07 | 76.35 | 46.25 | 69.06 | 64.99 |
| 50% | Palu | 6.02 | 40.89 | 8.72 | 43.80 | 73.34 | 76.17 | 72.98 | 42.24 | 67.32 | 62.64 |
| | **ReCalKV** | **5.83** | **39.51** | **8.14** | **45.00** | **74.39** | **76.39** | **74.71** | **43.52** | **67.80** | **63.64** |
| 60% | Palu | 6.81 | 51.32 | 10.69 | 40.00 | 68.55 | 74.10 | 68.99 | 38.14 | 63.38 | 58.85 |
| | **ReCalKV** | **6.21** | **65.73** | **8.95** | **41.40** | **72.36** | **76.17** | **72.73** | **41.13** | **68.03** | **61.97** |
| 70% | Palu | 8.62 | 83.19 | 15.01 | 34.20 | 59.30 | 68.82 | 57.87 | 31.66 | 61.01 | 52.14 |
| | **ReCalKV** | **6.75** | **75.78** | **10.05** | **39.80** | **69.59** | **74.48** | **70.37** | **39.42** | **65.75** | **59.90** |
| | | | | | **Mistral-7B-Instruct-v0.2** (Jiang et al., 2023) | | | | | | |
| 0% | Original | 5.94 | 32.46 | 9.72 | 46.80 | 83.68 | 80.41 | 81.31 | 55.63 | 74.35 | 70.36 |
| 50% | Palu | 6.33 | 37.38 | 10.79 | 44.60 | 80.89 | 79.33 | 79.21 | 54.27 | 73.40 | 68.62 |
| | **ReCalKV** | **6.30** | **38.12** | **10.73** | **44.40** | **81.06** | **80.20** | **80.05** | **54.86** | **73.56** | **69.02** |
| 60% | Palu | 7.07 | 49.45 | 12.93 | 42.80 | 75.30 | 77.26 | 74.07 | 50.00 | 70.72 | 65.03 |
| | **ReCalKV** | **6.81** | **47.27** | **11.98** | **44.00** | **77.88** | **79.27** | **77.78** | **52.22** | **72.30** | **67.24** |
| 70% | Palu | 8.71 | 77.51 | 16.78 | 38.60 | 66.48 | 75.24 | 66.96 | 42.49 | 66.54 | 59.39 |
| | **ReCalKV** | **8.08** | **70.96** | **14.98** | **39.00** | **71.08** | **76.82** | **72.14** | **46.25** | **67.72** | **62.17** |
| | | | | | **LongChat-7B-v1.5-32k** (Li et al., 2023) | | | | | | |
| 0% | Original | 7.61 | 89.04 | 10.52 | 41.40 | 71.28 | 76.12 | 71.84 | 41.38 | 68.19 | 61.70 |
| 50% | Palu | 8.11 | 120.11 | 12.08 | 38.20 | 68.30 | 72.52 | 67.93 | 38.40 | 64.96 | 58.39 |
| | **ReCalKV** | **7.89** | **95.51** | **11.48** | **41.80** | **69.66** | **73.78** | **69.61** | **38.91** | **65.59** | **59.89** |
| 60% | Palu | 9.15 | 168.94 | 14.42 | 37.80 | 64.76 | 69.70 | 60.14 | 34.64 | 61.09 | 54.68 |
| | **ReCalKV** | **8.14** | **108.52** | **12.12** | **40.00** | **67.63** | **71.98** | **66.92** | **37.03** | **63.77** | **57.89** |
| 70% | Palu | 11.95 | 172.23 | 20.87 | 32.20 | 54.94 | 64.74 | 50.00 | 28.67 | 55.56 | 47.69 |
| | **ReCalKV** | **9.01** | **109.38** | **13.63** | **35.20** | **63.18** | **68.55** | **58.84** | **33.53** | **59.12** | **53.07** |

## 4.2 Main Results

**Perplexity Results.** We evaluate the language modeling capability of ReCalKV on three standard datasets—WikiText2 (Merity et al., 2017), PTB (Plotz & Roth, 2017), and C4 (Raffel et al., 2020a)—using perplexity as the metric. As shown in Table 1, ReCalKV achieves lower perplexity than Palu (Chang et al., 2024) on most compression ratios and model families. On LLaMA-2-7B (Touvron et al., 2023b), ReCalKV yields a perplexity of 5.83 on WikiText2 and 8.14 on C4 at 50% compression, compared to 6.02 and 8.72 from Palu, respectively. Similar trends are observed on LLaMA-7B (Touvron et al., 2023a) and Mistral-7B (Jiang et al., 2023). Notably, on PTB, ReCalKV significantly outperforms Palu under aggressive compression. For instance, at 70% compression, the perplexity of Palu rises sharply to 211.33 on LLaMA-7B and 172.23 on LongChat-7B, while ReCalKV keeps it much lower at 71.47 and 109.38, respectively. These results demonstrate that ReCalKV maintains strong language modeling ability under high compression. Even at 70% compression, perplexity remains moderate, suggesting better information retention than low-rank baselines.

**Zero-shot Accuracy Results.** In addition to perplexity, we evaluate ReCalKV on six zero-shot QA datasets, including OBQA (Mihaylov et al., 2018), HellaSwag (Zellers et al., 2019a), PIQA (Bisk et al., 2020), ARC-e (Clark et al., 2018a), ARC-c (Clark et al., 2018a), and Winogrande (Sakaguchi et al., 2020). Across all model families and compression levels, ReCalKV demonstrates strong resilience in accuracy. While both methods see a decline in performance as the compression ratio increases, Palu (Chang et al., 2024) exhibits a significantly steeper drop. For instance, at 70% compression on LLaMA-2-7B (Touvron et al., 2023b), Palu's average accuracy drops to 52.14%,

Table 2: Evaluation results on LongBench (Bai et al., 2023), covering accuracy across 8 tasks and the overall average, comparing **ReCalKV** and Palu (Chang et al., 2024) under 50%–70% KV cache compression ratios.

| RATIO | METHOD | Qasper | QMSum | MultiNews | TREC | TriviaQA | SAMSum | LCC | RepoBench-P | Average↑ |
|---|---|---|---|---|---|---|---|---|---|---|
| | | | | **LLaMA-2-7B** (Touvron et al., 2023b) | | | | | | |
| 0% | Original | 9.58 | 21.22 | 3.51 | 66.00 | 87.72 | 41.66 | 66.68 | 59.80 | 44.52 |
| 50% | Palu | 8.40 | 18.93 | 1.31 | 61.50 | 84.56 | 38.40 | 50.90 | 46.80 | 38.85 |
| | **ReCalKV** | **8.39** | **18.89** | **1.37** | **58.50** | **84.75** | **39.41** | **58.29** | **54.61** | **40.53** |
| 60% | Palu | 5.10 | 16.51 | 2.13 | 55.50 | 59.84 | 33.13 | 29.62 | 33.56 | 29.42 |
| | **ReCalKV** | **6.62** | **17.96** | **0.17** | **58.00** | **80.41** | **38.13** | **49.05** | **44.43** | **36.85** |
| 70% | Palu | 4.54 | 9.99 | 1.40 | 39.00 | 16.98 | 19.18 | 1.75 | 7.52 | 13.26 |
| | **ReCalKV** | **3.28** | **15.41** | **0.12** | **53.00** | **66.24** | **32.61** | **34.11** | **32.15** | **29.62** |
| | | | | **LLaMA-2-13B-Chat** (Touvron et al., 2023b) | | | | | | |
| 0% | Original | 24.21 | 20.38 | 25.70 | 67.50 | 86.90 | 42.19 | 50.06 | 50.55 | 45.94 |
| 50% | Palu | 24.65 | 21.03 | 24.21 | 67.00 | 83.75 | 40.73 | 37.81 | 38.35 | 42.19 |
| | **ReCalKV** | **19.30** | **20.47** | **24.32** | **68.00** | **83.82** | **40.96** | **29.72** | **36.41** | **40.38** |
| 60% | Palu | 17.65 | 20.27 | 21.76 | 65.00 | 79.25 | 36.49 | 34.04 | 29.91 | 38.05 |
| | **ReCalKV** | **17.16** | **20.12** | **24.30** | **65.00** | **80.77** | **39.22** | **39.33** | **37.46** | **40.42** |
| 70% | Palu | 17.99 | 19.10 | 17.11 | 59.5 | 62.44 | 29.45 | 8.73 | 18.09 | 29.05 |
| | **ReCalKV** | **16.47** | **20.22** | **22.18** | **62.50** | **76.87** | **35.50** | **30.79** | **26.51** | **36.38** |
| | | | | **Mistral-7B-Instruct-v0.2** (Jiang et al., 2023) | | | | | | |
| 0% | Original | 32.51 | 24.29 | 26.96 | 71.00 | 86.23 | 42.95 | 55.89 | 54.12 | 49.24 |
| 50% | Palu | 31.16 | 23.76 | 25.82 | 69.50 | 83.12 | 39.15 | 42.01 | 45.18 | 44.96 |
| | **ReCalKV** | **31.71** | **23.35** | **26.43** | **70.00** | **82.51** | **39.22** | **46.12** | **45.21** | **45.57** |
| 60% | Palu | 21.21 | 23.73 | 24.75 | 68.00 | 76.59 | 36.14 | 26.24 | 30.48 | 38.39 |
| | **ReCalKV** | **24.98** | **24.26** | **25.32** | **71.00** | **75.53** | **37.42** | **34.28** | **36.79** | **41.19** |
| 70% | Palu | 6.59 | 21.35 | 17.84 | 61.00 | 44.73 | 28.06 | 15.05 | 21.87 | 27.06 |
| | **ReCalKV** | **9.13** | **23.01** | **20.85** | **65.00** | **51.44** | **31.37** | **15.13** | **22.66** | **29.82** |
| | | | | **LongChat-7B-v1.5-32k** (Li et al., 2023) | | | | | | |
| 0% | Original | 29.32 | 22.81 | 26.61 | 66.50 | 83.99 | 40.83 | 53.02 | 56.94 | 47.50 |
| 50% | Palu | 21.77 | 21.93 | 23.65 | 64.00 | 76.68 | 39.46 | 38.49 | 43.57 | 41.19 |
| | **ReCalKV** | **25.15** | **22.08** | **23.38** | **63.00** | **79.75** | **40.72** | **50.54** | **50.52** | **44.39** |
| 60% | Palu | 13.12 | 21.97 | 19.07 | 55.50 | 66.14 | 34.68 | 42.01 | 16.55 | 33.63 |
| | **ReCalKV** | **20.99** | **21.13** | **22.68** | **59.00** | **76.12** | **38.78** | **40.45** | **40.91** | **40.01** |
| 70% | Palu | 6.27 | 19.05 | 14.47 | 37.50 | 36.75 | 21.95 | 2.09 | 5.45 | 17.94 |
| | **ReCalKV** | **17.50** | **20.70** | **18.94** | **44.00** | **67.29** | **33.86** | **10.48** | **15.23** | **28.50** |

whereas ReCalKV retains 59.90%. Similar robustness is observed on Mistral-7B (Jiang et al., 2023) and LongChat-7B (Li et al., 2023), where ReCalKV consistently delivers higher or comparable average accuracy under the same compression levels. These results highlight ReCalKV's strong capability to preserve task performance even under aggressive KV cache size reductions. Moreover, its stability across diverse tasks highlights its practicality for efficient long-context inference.

**Longbench Result.** We further evaluate ReCalKV on LongBench (Bai et al., 2023), a benchmark designed to test long-context understanding across diverse tasks. As shown in Table 2, ReCalKV achieves higher average accuracy than Palu (Chang et al., 2024) across nearly all model scales and compression ratios. The gap becomes especially pronounced at high compression levels (e.g., 70%), where Palu suffers significant degradation while ReCalKV maintains competitive performance across all benchmarks. This underscores ReCalKV's robustness under memory constraints, enabling accurate and reliable long-context inference even at aggressive compression levels.

### 4.3 ABLATION STUDY

To analyze the individual contributions of each component in ReCalKV, we conduct ablation studies on LLaMA-2-7B (Touvron et al., 2023b) under a fixed 80% compression ratio. Table 3 reports perplexity on WikiText-2 (Merity et al., 2017), PTB (Plotz & Roth, 2017), and C4 (Raffel et al., 2020a), as well as accuracy on two downstream evaluation suites: the average accuracy over six zero-shot QA datasets (**zero-shot Avg. Acc**) and the average accuracy across eight tasks from the LongBench benchmark (Bai et al., 2023) (**LongBench Avg. Acc**).

**Ablation on Head-wise Similarity-aware Reordering (HSR).** By comparing the first and second rows in Table 3, we observe that enabling HSR alone (without offline calibration) significantly improves performance. For example, perplexity on WikiText-2 drops from 9.34 to 9.01, and Long-Bench accuracy increases from 9.01% to 12.44%. These results suggest that the reordering strategy

Table 3: Ablation studies on LLaMA-2-7B are conducted at a fixed 80% compression ratio. Perplexity is reported on WikiText-2, PTB, and C4, with our results in **bold**.

| HSR | OVC | WikiText2↓ | PTB↓ | C4↓ | zero-shot Avg. Acc↑ | LongBench Avg. Acc↑ |
|---|---|---|---|---|---|---|
| ✗ | ✗ | 9.34 | 92.52 | 14.58 | 49.01 | 9.01 |
| ✓ | ✗ | 9.01 | 87.58 | 14.16 | 52.33 | 12.44 |
| ✗ | ✓ | 8.91 | 81.96 | 14.08 | 52.98 | 13.09 |
| ✓ | ✓ | **8.48** | **79.04** | **13.29** | **54.55** | **15.40** |

in HSR effectively groups similar attention heads together before applying SVD, which reduces approximation error during low-rank decomposition and leads to improved model performance.

**Ablation on Offline Value Calibration (OVC).** Comparing the first and third rows, we assess the effect of offline calibration alone. Perplexity on WikiText-2 improves from 9.34 to 8.91, while LongBench accuracy rises to 13.09%. This confirms that calibrating the SVD decomposition of the Value projection matrix using a small held-out dataset effectively improves the quality of approximation, thereby enhancing model performance and robustness across tasks.

## 4.4 INTEGRATE WITH KV CACHE QUANTIZATION

To evaluate the compatibility of ReCalKV with quantization, we combine it with 4-bit and 3-bit per-token quantization under varying average ranks to simulate different compression ratios. We also apply a randomized Hadamard transform before quantization, following Palu (Chang et al., 2024), to improve robustness. As shown in Table 4, ReCalKV consistently outperforms Palu under the same bitwidth and compression settings. For instance, at 60% compression with 4-bit quantization, ReCalKV achieves 6.24 perplexity on WikiText2 (vs. 6.84 for Palu); at 70% with 3-bit, it reduces C4 perplexity from 15.75 to 10.41. These results highlight the synergy between ReCalKV and quantization for efficient KV cache compression.

Table 4: ReCalKV with KV cache quantization.

| RATIO | METHOD | BIT | WikiText-2↓ | C4↓ |
|---|---|---|---|---|
| 0% | Original | 16 | 5.47 | 7.26 |
| 50% | Palu | 4 | 6.04 | 8.75 |
| | Palu | 3 | 6.15 | 8.92 |
| | ReCalKV | 4 | **5.86** | **8.18** |
| | ReCalKV | 3 | **5.96** | **8.34** |
| 60% | Palu | 4 | 6.84 | 10.77 |
| | Palu | 3 | 7.01 | 11.06 |
| | ReCalKV | 4 | **6.24** | **9.01** |
| | ReCalKV | 3 | **6.39** | **9.21** |
| 70% | Palu | 4 | 8.71 | 15.17 |
| | Palu | 3 | 9.04 | 15.75 |
| | ReCalKV | 4 | **6.79** | **10.11** |
| | ReCalKV | 3 | **7.01** | **10.41** |

## 4.5 INFERENCE EFFICIENCY

To evaluate the practical runtime benefits of ReCalKV, we implement a custom fused attention kernel using Triton that integrates our low-rank compression for both Key and Value. For the Key path, we incorporate *Head-wise Similarity-aware Reordering* (HSR) as an online permutation step applied to each token during runtime. For the Value path, we perform *offline matrix fusion* to precompute and store a compact representation. The kernel supports rotary position embedding (RoPE) and is fully compatible with causal attention. We benchmark the latency of a single attention module on an NVIDIA A800 GPU across prompt lengths of 4K, 16K, and 65K, averaging over 100 runs per setting. As

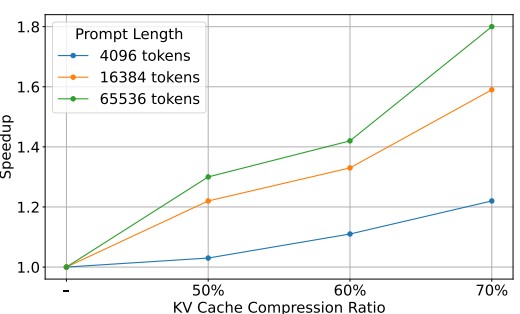

Figure 4: Latency speedup of **ReCalKV** relative to the baseline under various prompt lengths. Higher compression leads to greater acceleration, especially for longer prompts.

shown in Figure 4, ReCalKV achieves increasing latency speedups with higher KV compression and longer prompts, reaching up to 1.22×, 1.59×, and 1.80× improvements at 4K, 16K, and 65K respectively, under 70% compression. This trend confirms that our fused attention kernel becomes more effective as memory cost dominates, especially in long-context scenarios. These results validate the scalability and deployment efficiency of ReCalKV under strict memory budgets.

## 5 CONCLUSION

In this work, we propose ReCalKV, a post-training KV cache compression framework tailored for efficient long-context reasoning in LLMs. By exploiting the distinct characteristics of Keys and

Values in the attention mechanism, ReCalKV applies Head-wise Similarity-aware Reordering (HSR) and grouped SVD to compress Keys, while employing Offline Value Calibration (OVC) to compress Values. This design reduces hidden dimensions with minimal additional computation and preserves model performance under high compression ratios. Experimental results demonstrate that ReCalKV consistently outperforms existing low-rank compression methods, offering a practical and effective solution for memory-efficient LLM serving. Moreover, it can be combined with quantization to achieve higher compression with minimal performance loss. This work offers a promising direction for scalable and efficient deployment of long-context LLMs.

## ETHICS STATEMENT

The research conducted in the paper conforms, in every respect, with the ICLR Code of Ethics.

## REPRODUCIBILITY STATEMENT

We have provided implementation details in Section 4. We will also release all the code and models.

## LLM USAGE STATEMENT

Large Language Models (LLMs) were used solely for polishing writing. They did not contribute to the research content or scientific findings of this work.

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
