# ReCalKV: Low-Rank KV Cache Compression via Head Reordering and Offline Calibration

## CONTENTS

## A   DERIVATION OF OFFLINE CALIBRATION FORMULA

We aim to approximate the original weight matrix $\mathbf{W}$ by a low-rank decomposition $\mathbf{LR}$, such that the output difference with input $\mathbf{X}$ is minimized:

$$\mathcal{E} = \|\mathbf{LRX} - \mathbf{WX}\|_F^2. \tag{1}$$

We first take the partial derivative of $\mathcal{E}$ with respect to $\mathbf{L}$:

$$\frac{\partial \mathcal{E}}{\partial \mathbf{L}} = \frac{\partial}{\partial \mathbf{L}} \operatorname{Tr}\left((\mathbf{LRX} - \mathbf{WX})^\top (\mathbf{LRX} - \mathbf{WX})\right)$$
$$= 2\mathbf{LRXX}^\top \mathbf{R}^\top - 2\mathbf{WXX}^\top \mathbf{R}^\top. \tag{2}$$

Setting the gradient to zero yields the optimal solution:

$$\frac{\partial \mathcal{E}}{\partial \mathbf{L}} = 0 \quad \Rightarrow \quad \mathbf{L} = \mathbf{WXX}^\top \mathbf{R}^\top \left(\mathbf{RXX}^\top \mathbf{R}^\top\right)^{-1}. \tag{3}$$

Next, we take the partial derivative of $\mathcal{E}$ with respect to $\mathbf{R}$:

$$\frac{\partial \mathcal{E}}{\partial \mathbf{R}} = \frac{\partial}{\partial \mathbf{R}} \operatorname{Tr}\left(\mathbf{RXX}^\top \mathbf{R}^\top \mathbf{L}^\top \mathbf{L} - 2\mathbf{RXX}^\top \mathbf{W}^\top \mathbf{L}\right)$$
$$= 2\mathbf{RXX}^\top \mathbf{L}^\top \mathbf{L} - 2\mathbf{WXX}^\top \mathbf{L}. \tag{4}$$

Setting the gradient to zero gives:

$$\frac{\partial \mathcal{E}}{\partial \mathbf{R}} = 0 \quad \Rightarrow \quad \mathbf{R} = \left(\mathbf{L}^\top \mathbf{L}\right)^{-1} \mathbf{L}^\top \mathbf{W}. \tag{5}$$

## B   CENTERED KERNEL ALIGNMENT (CKA) SIMILARITY

Let the centered representations be denoted as

$$\widetilde{\mathbf{X}} = \mathbf{U_X}\boldsymbol{\Sigma_X}\mathbf{V_X}^\top, \quad \widetilde{\mathbf{Y}} = \mathbf{U_Y}\boldsymbol{\Sigma_Y}\mathbf{V_Y}^\top, \tag{6}$$

where $\mathbf{U_X}$ and $\mathbf{U_Y}$ are the left singular vectors of $\widetilde{\mathbf{X}}$ and $\widetilde{\mathbf{Y}}$, and $\boldsymbol{\Sigma_X}$, $\boldsymbol{\Sigma_Y}$ are their corresponding singular value matrices.

The centered Gram matrices are given by:

$$\widetilde{\mathbf{G}}_{\mathbf{X}} = \widetilde{\mathbf{X}}\widetilde{\mathbf{X}}^\top = \mathbf{U_X}\boldsymbol{\Sigma_X}^2\mathbf{U_X}^\top, \tag{7}$$
$$\widetilde{\mathbf{G}}_{\mathbf{Y}} = \widetilde{\mathbf{Y}}\widetilde{\mathbf{Y}}^\top = \mathbf{U_Y}\boldsymbol{\Sigma_Y}^2\mathbf{U_Y}^\top. \tag{8}$$

Then, the Hilbert-Schmidt Independence Criterion (HSIC) becomes:

$$\operatorname{HSIC}(\mathbf{X}, \mathbf{Y}) = \operatorname{Tr}(\widetilde{\mathbf{G}}_{\mathbf{X}}\widetilde{\mathbf{G}}_{\mathbf{Y}}) = \sum_{i=1}^{r_X}\sum_{j=1}^{r_Y} \sigma_{\mathbf{X},i}^2 \sigma_{\mathbf{Y},j}^2 \left(\mathbf{u}_{\mathbf{X}}^{(i)\top}\mathbf{u}_{\mathbf{Y}}^{(j)}\right)^2, \tag{9}$$

where $\mathbf{u}_{\mathbf{X}}^{(i)}$ and $\mathbf{u}_{\mathbf{Y}}^{(j)}$ denote the $i$-th and $j$-th columns of $\mathbf{U_X}$ and $\mathbf{U_Y}$, respectively.

The normalization terms are:

$$\operatorname{HSIC}(\mathbf{X}, \mathbf{X}) = \sum_{i=1}^{r_X} \sigma_{\mathbf{X},i}^4, \tag{10}$$
$$\operatorname{HSIC}(\mathbf{Y}, \mathbf{Y}) = \sum_{j=1}^{r_Y} \sigma_{\mathbf{Y},j}^4. \tag{11}$$

Hence, the final CKA similarity can be expressed as:

$$\operatorname{CKA}(\mathbf{X}, \mathbf{Y}) = \frac{\sum_{i=1}^{r_X}\sum_{j=1}^{r_Y} \sigma_{\mathbf{X},i}^2 \sigma_{\mathbf{Y},j}^2 \left(\mathbf{u}_{\mathbf{X}}^{(i)\top}\mathbf{u}_{\mathbf{Y}}^{(j)}\right)^2}{\sqrt{\left(\sum_{i=1}^{r_X} \sigma_{\mathbf{X},i}^4\right)\left(\sum_{j=1}^{r_Y} \sigma_{\mathbf{Y},j}^4\right)}}. \tag{12}$$

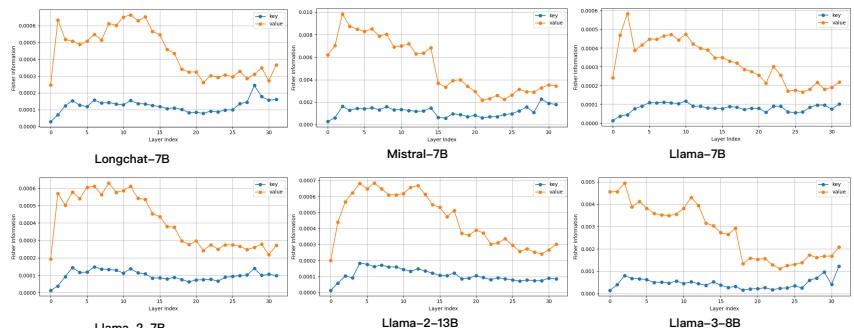

Figure 1: Comparison of Fisher information between Key and Value projection layers across different models.

This expression reveals that CKA measures the alignment between the principal subspaces of $\widetilde{\mathbf{X}}$ and $\widetilde{\mathbf{Y}}$. Larger CKA values indicate stronger overlap in dominant singular directions, as determined by both the magnitude of singular values and the cosine similarity between the corresponding left singular vectors.

## C  VISUALIZATION OF FISHER INFORMATION

We analyze the Fisher information of the Key and Value projection layers across different models. It can be observed that the Fisher information—i.e., the importance—of the Value projection layers is higher than that of the Key projection layers, as shown in Figure 1.

## D  THEORETICAL ANALYSIS OF COMPRESSION ERROR

To complement our empirical evaluations, we provide a rigorous theoretical analysis to justify the effectiveness of our SVD-based KV cache compression strategy. Specifically, we analyze the following three components: (i) general SVD approximation error bounds, (ii) error reduction via offline calibration of the Value projection, and (iii) Head-wise Similarity–aware Reordering (HSR) guided by Centered Kernel Alignment (CKA). These results offer interpretability and robustness guarantees for our proposed method.

### D.1  SVD APPROXIMATION ERROR BOUND

Let $W \in \mathbb{R}^{m \times n}$ be a weight matrix with full SVD:
$$W = U\Sigma V^\top,$$
where $U \in \mathbb{R}^{m \times m}$, $V \in \mathbb{R}^{n \times n}$ are orthogonal matrices, and $\Sigma = \mathrm{diag}(\sigma_1, \ldots, \sigma_n)$ contains the singular values in descending order:
$$\sigma_1 \geq \sigma_2 \geq \cdots \geq \sigma_n \geq 0.$$

Let $\widetilde{W} = U_k \Sigma_k V_k^\top$ be the rank-$k$ approximation using the top $k$ singular values and corresponding vectors. Then for any input vector $x \in \mathbb{R}^n$, the approximation error satisfies:
$$\|Wx - \widetilde{W}x\|_2 \leq \sigma_{k+1}\|x\|_2.$$

**Proof Sketch:** The error term can be rewritten as:
$$\|(W - \widetilde{W})x\|_2 = \|U(\Sigma - \Sigma_k)V^\top x\|_2 = \|(\Sigma - \Sigma_k)V^\top x\|_2.$$
Since $(\Sigma - \Sigma_k)$ zeroes out the top $k$ entries, we have:
$$\|(\Sigma - \Sigma_k)V^\top x\|_2^2 = \sum_{i=k+1}^{n} \sigma_i^2 (V^\top x)_i^2 \leq \sigma_{k+1}^2 \sum_{i=k+1}^{n} (V^\top x)_i^2 \leq \sigma_{k+1}^2 \|x\|_2^2.$$
Taking square roots yields the bound. When singular values decay rapidly—as empirically observed in transformer models—this guarantees low approximation error.

## D.2 OFFLINE CALIBRATION REDUCES VALUE COMPRESSION ERROR

Given a low-rank factorization of the Value matrix $W_v \approx L_v R_v$, we show that calibrating $(L_v, R_v)$ on a dataset $X$ guarantees lower or equal reconstruction error than vanilla SVD.

**Theorem 1.** Let $X \in \mathbb{R}^{n \times b}$ be a calibration dataset. Then:

$$\|L_v^* R_v^* X - W_v X\|_F^2 \le \|\tilde{L}_v \tilde{R}_v X - W_v X\|_F^2,$$

where $(\tilde{L}_v, \tilde{R}_v)$ are SVD factors, and $(L_v^*, R_v^*)$ are obtained via two-stage least-squares minimization.

**Proof Sketch:**

- Fix $\tilde{R}_v$ and solve for $L_v$:

$$L_v^* = W_v X (\tilde{R}_v X)^\top \left[ (\tilde{R}_v X)(\tilde{R}_v X)^\top \right]^{-1}$$

  This minimizes $\|L_v \tilde{R}_v X - W_v X\|_F^2$.
- Fix $L_v^*$ and solve for $R_v$:

$$R_v^* = (L_v^{*\top} L_v^*)^{-1} L_v^{*\top} W_v X X^\top (X X^\top)^{-1}$$

  This minimizes $\|L_v^* R_v X - W_v X\|_F^2$.
- By composition of two optimizations:

$$\|L_v^* R_v^* X - W_v X\|_F^2 \le \|\tilde{L}_v \tilde{R}_v X - W_v X\|_F^2.$$

This result shows that our offline calibration procedure is guaranteed to reduce (or at worst maintain) the approximation error relative to uncalibrated SVD.

## D.3 CKA SIMILARITY IMPLIES BETTER SVD APPROXIMATION (JUSTIFYING HSR)

We provide theoretical justification for our Head-wise Similarity–aware Reordering (HSR) by showing that higher CKA similarity between adjacent heads leads to better low-rank approximation after concatenation.

**Theorem 2.** Let $A_1 \in \mathbb{R}^{m \times d_1}$, $A_2 \in \mathbb{R}^{m \times d_2}$ and $W_k = [A_1, A_2] \in \mathbb{R}^{m \times (d_1 + d_2)}$. If $\mathrm{CKA}(A_1, A_2)$ increases, the truncated SVD error:

$$E_r := \left\| W_k - \sum_{i=1}^r \sigma_i u_i v_i^\top \right\|_F^2 = \sum_{i=r+1}^{\mathrm{rank}(W_k)} \sigma_i^2$$

decreases.

**Proof Sketch:**

- The Gram matrix is:

$$W_k W_k^\top = G_1 + G_2 = A_1 A_1^\top + A_2 A_2^\top$$

  The eigenvalues of $G_1 + G_2$ determine the singular values of $W_k$, hence the tail sum of eigenvalues $\lambda_{r+1}, \ldots$ controls the SVD error.
- Define centered Gram matrices:

$$\tilde{G}_1 = \tilde{A}_1 \tilde{A}_1^\top, \quad \tilde{G}_2 = \tilde{A}_2 \tilde{A}_2^\top$$

  with $\tilde{A}_i = H A_i$, $H = I_m - \frac{1}{m} \mathbf{1} \mathbf{1}^\top$.
- CKA is:

$$\mathrm{CKA}(A_1, A_2) = \frac{\mathrm{tr}(\tilde{G}_1 \tilde{G}_2)}{\|\tilde{G}_1\|_F \cdot \|\tilde{G}_2\|_F}$$

- Higher CKA implies the dominant eigenspaces of $G_1$ and $G_2$ align. Therefore, their sum $G_1 + G_2$ concentrates spectral energy in the top $r$ components, reducing $\sum_{i=r+1}^{\mathrm{rank}} \lambda_i$.

Thus, HSR—which reorders heads by maximizing local CKA similarity—effectively reduces the approximation error under truncated SVD.

### D.4 SUMMARY

We have provided theoretical justification for key components of ReCalKV:

- Low-rank SVD has a provable bound on approximation error proportional to $\sigma_{k+1}$.
- Offline calibration yields lower reconstruction error than uncalibrated SVD.
- HSR reduces SVD error by aligning similar head structures with high CKA.

These results support the reliability and effectiveness of our proposed KV cache compression pipeline.

## E  CALIBRATION SENSITIVITY ANALYSIS

To assess the robustness of ReCalKV to the calibration process, we investigate how different calibration datasets and calibration set sizes affect post-compression model quality. Specifically, we evaluate the perplexity of LLaMA-2-7B compressed with ReCalKV under varying calibration configurations.

**Experimental Setup.** We consider three types of calibration datasets: WikiText2, C4, and Penn Treebank (PTB). For each dataset, we sample subsets of size {16, 32, 64, 128, 256} as calibration data. The downstream evaluation is performed on all three datasets (WikiText2, C4, PTB) to comprehensively assess generalization across domains.

**Results and Observations.** Table 1 presents the perplexity results across all calibration dataset types and sizes. Several consistent trends emerge from the analysis:

- **Calibration Dataset Type.** When the calibration size is fixed (e.g., 256), the choice of dataset (WikiText2, C4, or PTB) has minimal impact on the final performance. For instance, with 256 calibration samples:
  - WikiText2 calibration yields PPL = 5.83 (WikiText2), 8.14 (C4), 39.51 (PTB)
  - C4 calibration yields PPL = 5.88 (WikiText2), 8.16 (C4), 40.41 (PTB)
  - PTB calibration yields PPL = 5.88 (WikiText2), 8.20 (C4), 40.04 (PTB)

  This suggests that the method is agnostic to the choice of calibration dataset, as long as it reflects general language statistics.

- **Calibration Set Size.** When fewer than 32 calibration samples are used, performance degradation becomes more significant. For example, with WikiText2 calibration:
  - 256 samples: 5.83 (WikiText2), 8.14 (C4), 39.51 (PTB)
  - 32 samples: 5.86, 8.21, 40.53
  - 16 samples: 6.32, 8.92, 46.29

  This trend holds across all calibration datasets, indicating that while ReCalKV remains stable when the calibration size is greater than or equal to 32, extremely small sample sizes (e.g., 16) may be insufficient to capture representative statistical patterns.

**Conclusion.** These findings confirm that ReCalKV is highly robust to the choice of calibration dataset and maintains stable performance even with modest calibration sizes. For reliable compression quality, using 32 or more calibration samples is sufficient in most practical scenarios.

## F  CALIBRATION SETUP AND RESOURCE OVERHEAD

The proposed method employs an offline calibration strategy rather than additional training or fine-tuning. This calibration process is lightweight and fully feedforward, without any backpropagation or parameter updates.

**Calibration Procedure.** Following common practice in post-training quantization and compression, the calibration dataset is drawn from WikiText2. Only 256 samples are used to calibrate the Key and Value projection matrices. This small-scale calibration is sufficient to capture representative activation statistics while keeping overhead minimal.

Table 1: Perplexity results of LLaMA-2-7B after compression with ReCalKV, under varying calibration datasets and sizes.

| Calibration Data | Data Size | WikiText2 PPL ↓ | C4 PPL ↓ | PTB PPL ↓ |
|---|---|---|---|---|
| WikiText2 | 256 | 5.83 | 8.14 | 39.51 |
| WikiText2 | 128 | 5.86 | 8.20 | 40.37 |
| WikiText2 | 64 | 5.86 | 8.20 | 40.42 |
| WikiText2 | 32 | 5.86 | 8.21 | 40.53 |
| WikiText2 | 16 | 6.32 | 8.92 | 46.29 |
| C4 | 256 | 5.88 | 8.16 | 40.41 |
| C4 | 128 | 5.89 | 8.17 | 40.38 |
| C4 | 64 | 5.89 | 8.17 | 40.38 |
| C4 | 32 | 5.90 | 8.17 | 40.47 |
| C4 | 16 | 6.66 | 9.73 | 73.86 |
| PTB | 256 | 5.88 | 8.20 | 40.04 |
| PTB | 128 | 5.88 | 8.20 | 40.06 |
| PTB | 64 | 5.89 | 8.20 | 40.05 |
| PTB | 32 | 5.89 | 8.21 | 40.03 |
| PTB | 16 | 6.14 | 8.52 | 41.17 |

**Resource Consumption.** Table 2 reports the calibration time and GPU memory usage for two representative models: LLaMA-2-7B and LLaMA-2-13B-chat. All measurements are taken on a single NVIDIA A6000 GPU.

Table 2: Calibration time and GPU memory usage.

| Model | Calibration Time | GPU Memory |
|---|---|---|
| LLaMA-2-7B | 11 minutes | 25 GB |
| LLaMA-2-13B-chat | 25 minutes | 34 GB |

**Practical Implications.** As shown, the entire calibration process completes in under 30 minutes, even for models with 13 billion parameters. The GPU memory footprint remains moderate, making the method accessible on widely available hardware. These properties suggest that the proposed compression strategy is efficient, scalable, and suitable for real-world deployment in both research and production settings.

## G ADDITIONAL EXPERIMENTS

To further validate the effectiveness and generalizability of ReCalKV, we conduct additional experiments from two perspectives: (i) comparison with more baseline methods, and (ii) evaluation on newer large language models (LLMs).

### G.1 COMPARISON WITH ADDITIONAL BASELINES

We compare ReCalKV with two representative methods: Eigen Attention (Saxena et al., 2024) and ASVD Yuan et al. (2023). These baselines represent different lines of compression strategies, providing a comprehensive comparison.

**(a) Comparison with Eigen Attention.** Table 3 shows the perplexity of ReCalKV and Eigen Attention on LLaMA-2-7B under various compression ratios. ReCalKV consistently achieves lower perplexity on both WikiText2 and C4, demonstrating superior retention of contextual information under compression.

**(b) Comparison with ASVD.** To further assess robustness across compression levels, we compare ReCalKV with ASVD on LLaMA-2-13B. As shown in Table 4, ReCalKV outperforms ASVD at

Table 3: Comparison with Eigen Attention on LLaMA-2-7B. Lower perplexity is better.

| Method | Compression Ratio | WikiText2 ↓ | C4 ↓ |
|---|---|---|---|
| Eigen Attention | 20% | 5.96 | 7.82 |
| ReCalKV | 20% | **5.53** | **7.39** |
| Eigen Attention | 30% | 6.28 | 8.55 |
| ReCalKV | 30% | **5.57** | **7.52** |
| Eigen Attention | 40% | 7.48 | 10.07 |
| ReCalKV | 40% | **5.67** | **7.74** |

all compression ratios except for the extreme 20%, where its perplexity is marginally higher. These results confirm that ReCalKV provides stable and high-fidelity compression even under aggressive constraints.

Table 4: Comparison with ASVD on LLaMA-2-13B under different compression ratios (CR). Metric: WikiText2 PPL.

| Method | 90% | 80% | 70% | 60% | 50% | 40% | 30% | 20% |
|---|---|---|---|---|---|---|---|---|
| ASVD | 4.89 | 4.90 | 4.91 | 4.92 | 4.96 | 5.08 | 5.33 | **6.06** |
| ReCalKV | 4.89 | **4.89** | **4.90** | **4.91** | **4.95** | **5.02** | **5.19** | 6.19 |

## G.2 EVALUATION ON NEWER MODELS

To demonstrate generalization to more recent LLMs, we evaluate ReCalKV on LLaMA-3.1-8B and compare it with Palu Chang et al. (2024), a state-of-the-art low-rank compression method. Table 5 reports results under 50%–70% compression. ReCalKV consistently surpasses Palu in perplexity on both WikiText2 and C4, confirming its adaptability to newer architectures.

Table 5: Evaluation of ReCalKV vs. Palu on LLaMA-3.1-8B under various compression ratios.

| Model | Method | Compression | WikiText2 ↓ | C4 ↓ |
|---|---|---|---|---|
| | FP16 | 0% | 6.24 | 9.54 |
| | Palu | 50% | 8.71 | 19.02 |
| | ReCalKV | 50% | **8.59** | **18.04** |
| LLaMA-3.1-8B | Palu | 60% | 10.98 | 24.55 |
| | ReCalKV | 60% | **10.60** | **24.49** |
| | Palu | 70% | 16.99 | 41.86 |
| | ReCalKV | 70% | **15.81** | **39.34** |

These results verify that ReCalKV maintains strong compression performance on modern model architectures without requiring retraining.

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

# H  ADDITIONAL ANALYSIS ON RoPE-INDUCED RECONSTRUCTION OVERHEAD

Most modern LLMs (e.g., LLaMA, Mistral, Qwen; Touvron et al. (2023); **?**); **?**) adopt rotary positional embeddings (RoPE), which are applied *after* the linear Key projection. This creates a fundamental constraint for low-rank Key compression: because RoPE is defined only in the original $d_{\text{model}}$ (or head) dimensional space, it *cannot* be applied in the low-rank $r$-dimensional subspace. As a result, any compressed Key representation must be fully reconstructed before RoPE can be applied.

Formally, given

$$\mathbf{K} = \mathbf{X}\mathbf{W}_K \in \mathbb{R}^{T \times d}, \qquad \mathbf{Z} = \mathbf{X}\mathbf{L} \in \mathbb{R}^{T \times r},$$

the attention mechanism requires the reconstructed and position-encoded Key:

$$\widetilde{\mathbf{K}} = \text{RoPE}(\mathbf{Z}\mathbf{R}) \in \mathbb{R}^{T \times d}.$$

Thus the reconstruction $\mathbf{Z}\mathbf{R}$ is *mandatory*, regardless of the compression ratio $r/d$.

**Computational Overhead.**  Let $T$ be the sequence length and $\rho = r/d$ the compression ratio. The reconstruction requires

$$\mathcal{O}(T \cdot r \cdot d) = \mathcal{O}(Td^2\rho),$$

which grows linearly with $T$. This makes the overhead negligible for short prompts but dominant at long contexts (e.g., 32K–128K tokens). Moreover, since RoPE applies a complex rotation in the full-dimensional $d$ space, Key reconstruction is inherently more expensive than Value compression, which can be performed entirely within the reduced $r$-dimensional space.

**Implication.**  Since Key compression must always incur this unavoidable reconstruction cost, practical Key compression methods must jointly optimize memory savings, reconstruction complexity, and accuracy. This motivates our design in ReCalKV: we reduce unnecessary reconstruction through similarity-aware head reordering (HSR), improve reconstruction stability through whitening and calibration, and combine low-rank compression with quantization to mitigate the RoPE-induced overhead.

In summary, RoPE makes Key compression fundamentally more expensive than Value compression, and analyzing this trade-off is crucial for designing efficient KV cache compression algorithms.