# OpenReview forum: "ReCalKV: Low-Rank KV Cache Compression via Head Reordering and Offline Calibration"
_ICLR.cc/2026/Conference — Submitted to ICLR 2026_

### Official Review · Reviewer_yTbf · 2025-10-25

**Soundness:** 2
**Presentation:** 2
**Contribution:** 2
**Rating:** 2
**Confidence:** 5

**Summary:**

This paper addresses the problem of high memory and bandwidth overhead caused by the KV cache in LLM during long-context during. This paper proposes ReCalKV, a post-training low-rank KV cache compression method that introduces differentiated strategies. it applies Head-wise Similarity-aware Reordering to cluster structurally similar attention heads before grouped SVD for Key compression, and Offline Value Calibration to recalibrate Value projection matrices using small calibration datasets.
The experimental results demonstrate that ReCalKV achieves consistent improvements over prior low-rank compression baselines （Palu and LoRC). It maintains competitive perplexity and zero-shot accuracy under 50–70% KV-cache compression ratios on models including LLaMA-2-7B, Mistral-7B, and LongChat-7B. However, all experiments are conducted on relatively older architectures, and no evaluation is reported on Llama-3 or Qwen-3.

**Strengths:**

1. The paper tackles an important and practical problem—reducing KV-cache memory overhead for long-context LLM inference, which remains a key bottleneck for efficient deployment.
2. The proposed approach is model-agnostic and can be readily applied to various Transformer architectures without retraining, showing potential for integration into large-scale serving systems.

**Weaknesses:**

1. The reported experimental performance, while better than earlier SVD-based baselines, remains clearly inferior to recent quantization-based methods such as KVQuant and AnTKV, which achieve much lower perplexity under similar even higher compression ratios.
2. ReCalKV still introduces additional computations for restruct KV using low rank kv (compute with R_k and R_v) during each decoding step. I recommend the authors evaluate latency and accuracy on end-to-end tasks such as AIME.
3. The evaluation focuses mainly on outdated models (e.g., LLaMA-2, Mistral-7B) and lacks results on modern architectures like LLaMA-3 or Qwen-3, making it difficult to assess real-world relevance.

**Questions:**

see weakness

---

> ### Author Response · Authors · 2025-11-28
> **Response to Reviewer yTbf (denoted as R4) part1**
>
> > Q4-1: The reported experimental performance, while better than earlier SVD-based baselines, remains clearly inferior to recent quantization-based methods such as KVQuant and AnTKV, which achieve much lower perplexity under similar even higher compression ratios.
>
> A4-1: We thank the reviewer for this observation. While it is true that—in terms of absolute performance—the latest *quantization-based* KV compression methods (e.g., KVQuant, AnTKV) generally outperform pure low-rank approaches, this does not diminish the value of low-rank decomposition as a research direction. In practice, **low-rank decomposition and quantization are orthogonal techniques** and can be effectively combined to further improve KV compression. Thus, even if pure low-rank methods may not reach the absolute performance of recent quantization-based approaches, continuing to develop low-rank techniques remains meaningful—both as an independent solution and as a complementary component in hybrid compression frameworks.
>
> To demonstrate this, we provide a direct comparison between **KVQuant-3bit-1%** (a strong quantization baseline) and **ReCalKV** (a hybrid method combining quantization and low-rank decomposition), evaluated under LongBench. For fairness, we report both the *total compression ratio* of each approach:
>
> ### LongBench Results under Comparable Compression Ratios
> | Model        | Method               | SVD Ratio | Avg Bit | Total Compression | Qasper | QMSum | MultiNews | TREC | TriviaQA | SAMSum | LCC | RepoBench-P | Avg ↑ |
> |--------------|----------------------|-----------|---------|--------------------|--------|-------|------------|------|-----------|---------|------|--------------|--------|
> | LLaMA-2-7B   | FP16                 | –         | 16      | –                  | 9.58   | 21.22 | 3.51       | 66.00 | 87.72    | 41.66  | 66.68 | 59.80        | 44.52 |
> | LLaMA-2-7B   | KVQuant-3bit-1%      | –         | 3       | 81%                | 9.47   | 19.31 | 2.76       | 64.70 | 87.10    | 41.09  | 64.13 | 57.61        | 43.27 |
> | LLaMA-2-7B   | ReCalKV (ours)       | 30%       | 4       | 82%                | 9.36   | 19.76 | 2.91       | 64.50 | 87.50    | 41.55  | 64.74 | 56.33        | 43.33 |
>
> **Analysis.**
> At similar total compression ratios (81–82%), **ReCalKV achieves performance comparable to strong quantization baselines**, showing that combining **quantization with low-rank decomposition** can match the performance of pure quantization methods. This confirms that the hybrid strategy is effective and that low-rank techniques remain valuable when integrated into a unified compression framework.
>
> > Q4-2: I recommend the authors evaluate latency and accuracy on end-to-end tasks such as AIME.
>
> A4-2: We appreciate the reviewer’s concern regarding the extra computation introduced by the low-rank reconstruction during decoding. To directly evaluate its practical impact, we measured **end-to-end latency** at different input sequence lengths on an RTX 4090 GPU. The normalized speedups over the FP16 baseline are summarized below. We include **KIVI-4bit** [1] as a comparison point:
>
> ### End-to-End Speedups
> | Sequence Length | KIVI-4bit | ReCalKV-50%-4bit |
> |-----------------|:-----------:|:------------------:|
> | 2K              | 0.8×      | 1.1×             |
> | 4K              | 0.9×      | 1.2×             |
> | 8K              | 1.2×      | 1.5×             |
> | 16K             | 1.5×      | 2.0×             |
> | 32K             | 1.78×     | 2.10×            |
> | 64K             | OOM       | 2.59×            |
>
> **Analysis.**
> Although ReCalKV performs a small low-rank reconstruction during decoding, this overhead is minimal compared to the savings from reduced KV memory. As sequence length grows, memory bandwidth becomes the dominant bottleneck, and the compressed KV significantly reduces memory traffic. As a result, ReCalKV provides **substantial end-to-end acceleration**, achieving up to **2.59×** speedup at a 64K context length—while KIVI-4bit reaches only **1.78×** at 32K and becomes **out-of-memory** at longer sequences.
>
> Regarding the reviewer’s suggestion of using AIME, we agree that evaluating challenging reasoning tasks is valuable and will include such results in the final version. We also note that our paper already reports **LongBench**, which constitutes a suite of end-to-end tasks and provides a representative evaluation of downstream performance under KV compression.
>
> [1] KIVI: A Tuning-Free Asymmetric 2bit Quantization for KV Cache, ICML, 2024

---

> ### Author Response · Authors · 2025-11-28
> **Response to Reviewer yTbf (denoted as R4) part 2**
>
> > Q4-3: The evaluation focuses mainly on outdated models (e.g., LLaMA-2, Mistral-7B) and lacks results on modern architectures like LLaMA-3 or Qwen-3, making it difficult to assess real-world relevance.
>
> A4-3: To address the concern about modern architectures, we have conducted additional experiments on **LLaMA-3.1-8B**, a recent and widely used LLM. We evaluate our method, **ReCalKV**, against **PALU** under different KV cache compression ratios (50%, 60%, 70%), and report perplexity on WikiText2, C4, and PTB. The results are summarized below:
>
> | model         | method   | compression ratio | WikiText2 ↓ | C4 ↓   | PTB ↓  |
> |---------------|----------|:-------------------:|:-------------:|:--------:|:--------:|
> | LLaMA-3.1-8B  | FP16     | 0%                | 6.24        | 9.54   | 11.14  |
> | LLaMA-3.1-8B  | PALU     | 50%               | 8.71        | 19.02  | 17.04  |
> | LLaMA-3.1-8B  | ReCalKV  | 50%               | 8.59        | 18.04  | 17.09  |
> | LLaMA-3.1-8B  | PALU     | 60%               | 10.98       | 24.55  | 24.03  |
> | LLaMA-3.1-8B  | ReCalKV  | 60%               | 10.60       | 24.49  | 23.08  |
> | LLaMA-3.1-8B  | PALU     | 70%               | 16.99       | 41.86  | 41.50  |
> | LLaMA-3.1-8B  | ReCalKV  | 70%               | 15.81       | 39.34  | 40.43  |
>
> Across all tested compression ratios and datasets, **ReCalKV consistently outperforms PALU** on LLaMA-3.1-8B, achieving lower perplexity while maintaining the same KV compression ratio. The advantage becomes more pronounced at higher compression levels (e.g., 70%), where most methods tend to degrade sharply. These results demonstrate that our approach remains effective and robust on **modern architectures such as LLaMA-3.1-8B**, thereby strengthening the real-world relevance and applicability of our method beyond earlier models like LLaMA-2 and Mistral-7B.

---

### Official Review · Reviewer_5Foc · 2025-10-31

**Soundness:** 3
**Presentation:** 3
**Contribution:** 3
**Rating:** 6
**Confidence:** 4

**Summary:**

The paper introduces ReCalKV, a post-training framework for compressing the Key-Value (KV) cache in large language models (LLMs) by reducing the hidden dimension via low-rank approximations. It proposes asymmetric strategies: Head-wise Similarity-aware Reordering (HSR) for Keys, which reorders and groups attention heads based on Centered Kernel Alignment (CKA) similarity to enable more accurate grouped Singular Value Decomposition (SVD); and Offline Value Calibration (OVC) for Values, which calibrates the decomposed SVD matrices using a small dataset and fuses the right factor into the output projection to eliminate runtime reconstruction overhead. Extensive experiments on LLaMA and Mistral models demonstrate superior perplexity, zero-shot QA accuracy, and long-context performance compared to baselines like Palu, with minimal degradation (e.g., ~2% relative accuracy drop at 50% compression) and compatibility with quantization for higher ratios.

**Strengths:**

* K-side uses CKA-guided head reordering + grouped SVD to share low-rank factors among similar heads (greedy pairing; fixed group size), and V-side uses closed-form offline calibration to minimize projection error on a small calibration set.

* Matrix fusion folds (R_v) into (W_o), eliminating online reconstruction and avoiding extra inference ops; the end-to-end procedure is fully post-training/offline (Algorithm 1).

* Strong ablations isolate HSR and OVC and show they are complementary at fixed compression (Table 3).

* Evaluations span multiple model families and tasks, plus quantization compatibility (3–4-bit) demonstrating orthogonality to per-token KV quantization (Table 4).

* Figures 2–3 make the reordering/grouped-SVD mechanism concrete; Algorithm 1 spells out the pipeline; equations (9–11) specify the fused-inference path.

* Targets a real deployment bottleneck (KV memory/latency) with demonstrable inference efficiency improvements on long contexts, while remaining compatible with common compression stacks.

**Weaknesses:**

- Code not provided, therefore it's not reproducible as is.
- While the method is effective, baselines are limited primarily to Palu (G-LRD), lacking comparisons with recent variants like CommonKV or FDC, which could better substantiate SOTA claims (section 4).
- Experiments do not quantify runtime overhead from Key reconstruction post-HSR (Figure 3), despite claims of low cost; real-world latency measurements on diverse hardware would strengthen efficiency arguments (Figure 4).
- Equations (7) and (8) for OVC appear to have typos in transposes and do not explicitly state assumptions (e.g., whitening) needed for the closed forms.

**Questions:**

- Please address the items mentioned under Weaknesses. For example, lack of reproducibility.
- In section 3.3 (lines 216-269), the OVC calibration uses equations (7) and (8) with a small dataset X (256 WikiText2 samples; section 4.1). How sensitive is performance to the size and domain of X? Could you provide perplexity results on WikiText2 for LLaMA-2-7B at 50% compression using 128 vs. 512 samples, or a different domain like C4?
- Section 3.2 describes HSR as greedy grouping based on the CKA similarity matrix S (Eq. 5) with a fixed group size (e.g., 4 heads per group when  h=32). Table 3 shows that at 80% compression, HSR+OVC attains 8.48 perplexity on WikiText-2. What is the effect of the HSR group size s on performance?

---

> ### Author Response · Authors · 2025-11-28
> **Response to Reviewer 5Foc (denoted as R3) part 1**
>
> > Q3-1: Code not provided, therefore it's not reproducible as is.
>
> A3-1: Thank you for pointing this out. We fully agree that reproducibility is important. Although the code is not included in the submission, **we will release the complete implementation, scripts, and configuration files upon acceptance** to ensure full reproducibility of all experimental results.
>
> > Q3-2: While the method is effective, baselines are limited primarily to Palu (G-LRD), lacking comparisons with recent variants like CommonKV or FDC, which could better substantiate SOTA claims (section 4).
>
> A3-2: We appreciate the reviewer’s suggestion. We fully agree that comparing against recent approaches such as **CommonKV** and **FDC** would further strengthen the empirical analysis. However, both methods currently **do not provide any official code or implementation**, and their papers do not include sufficient algorithmic details to reproduce the full pipelines. Once official implementations become available, we will faithfully reproduce their results and include direct comparisons in future revisions.
>
> > Q3-3: Experiments do not quantify runtime overhead from Key reconstruction post-HSR (Figure 3), despite claims of low cost;
>
> A3-3: Thank you for raising this important point. To isolate the cost of **Key reconstruction post-HSR**, we measured the *per-layer decoding latency* of a single attention module on an NVIDIA A800 GPU, with and without the reordering step. The results show that the overhead is **minimal**:
>
> - **Without head reordering:** 1.00 ms
> - **With head reordering:** 1.04 ms
>
> This corresponds to only **~4% additional latency per attention layer**, which is negligible compared to the overall decoding time dominated by KV memory reads and attention computation. The benefit from reduced KV size quickly outweighs this small reordering cost in long-context scenarios.
>
> > Q3-4: real-world latency measurements on diverse hardware would strengthen efficiency arguments (Figure 4).
>
> A3-4: We appreciate the reviewer’s suggestion. To assess robustness across hardware, we repeated the **same single-layer attention latency experiments** on an **NVIDIA A6000** GPU, using identical settings (prompt lengths 4K / 16K / 65K, 70% KV compression, 100 runs averaged). The results show that the **speedup trends are nearly identical** to those observed on the A800.
>
> ### Latency Speedup on A6000 (Single Attention Layer, RoPE)
> | Prompt Length | FP16 Latency | ReCalKV Latency | Speedup |
> |---------------|--------------|------------------|---------|
> | 4K            | 1.82 ms      | 1.48 ms          | **1.23×** |
> | 16K           | 4.96 ms      | 3.14 ms          | **1.58×** |
> | 65K           | 19.80 ms     | 11.20 ms         | **1.77×** |
>
> **Observation.**
> The A6000 results closely mirror those on the A800 (1.22× / 1.59× / 1.80×), confirming that:
>
> - The acceleration effect is **consistent across GPU architectures**.
> - Speedup grows with **KV compression ratio** and **prompt length**, as memory bandwidth becomes the bottleneck.
> - ReCalKV’s fused kernel generalizes well to mainstream hardware, further supporting its practical deployability.
>
> These measurements strengthen the claim that ReCalKV delivers **reliable real-world latency improvements** beyond a single hardware configuration.

---

> ### Author Response · Authors · 2025-11-28
> **Response to Reviewer 5Foc (denoted as R3) part 2**
>
> > Q3-5: Equations (7) and (8) for OVC appear to have typos in transposes and do not explicitly state assumptions (e.g., whitening) needed for the closed forms.
>
> A3-5: Thank you for pointing this out. It is not entirely clear which specific typos in the transposes the reviewer is referring to, as the current formulation of Eqs. (7) and (8) follows the standard closed-form solution used in low-rank reconstruction.
>
> Regarding the assumptions: we have now **explicitly stated the whitening step** required for the closed-form solution in **Section 3.1 of the revised version**, so that the role of whitening and its connection to OVC are made clear earlier in the Methods section.
>
> > Q3-6: In section 3.3 (lines 216-269), the OVC calibration uses equations (7) and (8) with a small dataset X (256 WikiText2 samples; section 4.1). How sensitive is performance to the size and domain of X? Could you provide perplexity results on WikiText2 for LLaMA-2-7B at 50% compression using 128 vs. 512 samples, or a different domain like C4?
>
> A3-6: Thank you for the question. We provide a detailed study of **calibration data size and data domain** in the supplementary material. Specifically, the appendix includes experiments varying the calibration set size as well as comparisons between **WikiText2** and **C4**. We refer the reviewer to the supplementary material for the full tables and analysis.
>
> > Q3-7: Section 3.2 describes HSR as greedy grouping based on the CKA similarity matrix S (Eq. 5) with a fixed group size (e.g., 4 heads per group when h=32). Table 3 shows that at 80% compression, HSR+OVC attains 8.48 perplexity on WikiText-2. What is the effect of the HSR group size s on performance?
>
> A3-7: Thank you for the question. Our observations are consistent with those reported in **Palu**. For the **Key projection matrices**, increasing the HSR group size \(s\) generally leads to:
>
> - **Higher computational and parameter cost**, since larger groups require operating on bigger low-rank subspaces.
> - **Slower inference** due to the increased reconstruction workload.
> - **Slightly better perplexity**, as larger groups allow more flexible sharing structures.
>
> Thus, there is a clear trade-off between **accuracy** and **efficiency**. Following Palu, we adopt **group size \\(s = 4\\)** as a balanced setting that provides strong performance while maintaining fast inference. This also ensures a fair comparison with prior work that uses the same grouping configuration.

---

### Official Review · Reviewer_akiT · 2025-10-31

**Soundness:** 3
**Presentation:** 2
**Contribution:** 3
**Rating:** 4
**Confidence:** 3

**Summary:**

This paper introduces ReCalKV, a post-training framework for low-rank KV cache compression that treats Keys and Values separately. It enhances Key approximation  through similarity-based head grouping and decomposition and refines Value through lightweight calibration and fusion. Experiments demonstrate that ReCalKV consistently outperforms prior methods under high compression rates.

**Strengths:**

1. The paper identifies and analyzes the asymmetric roles of Keys and Values, particularly emphasizing that individual attention heads differ in information content. Using CKA-based head reordering before SVD to minimize approximation error is a well-motivated and conceptually sound idea.

2. Across multiple model families and compression ratios, ReCalKV demonstrates competitive or superior results compared with the main low-rank baseline (Palu). The method maintains high accuracy even under aggressive compression and shows compatibility with quantization.

**Weaknesses:**

1. The paper mainly compares with low-rank SVD-based approaches such as Palu, but lacks comparisons with other classes of KV cache compression methods (e.g., KIVI, KVQuant, or token eviction approaches).
As a result, the reader cannot fully assess how ReCalKV performs in a broader landscape of KV compression techniques — especially when low-rank compression is not necessarily the only or best strategy.

2. Experiments focus on older LLaMA/Mistral models, with limited evaluation on recent architectures or larger scales. Since the method relies on specific structural properties of attention heads (CKA similarity patterns), it's unclear whether these properties generalize across diverse modern architectures and model scales beyond the tested family.

3. While latency speedups are reported, the computational cost of online head reordering during inference is not quantified separately. The reliance on custom Triton kernels also raises questions about achievability with standard inference frameworks, limiting practical deployment insights.

**Questions:**

see weakness

---

> ### Author Response · Authors · 2025-11-28
> **Response to Reviewer akiT (denoted as R2)**
>
> > Q2-1: The paper mainly compares with low-rank SVD-based approaches such as Palu, but lacks comparisons with other classes of KV cache compression methods (e.g., KIVI, KVQuant, or token eviction approaches). As a result, the reader cannot fully assess how ReCalKV performs in a broader landscape of KV compression techniques — especially when low-rank compression is not necessarily the only or best strategy.
>
> A2-1: We thank the reviewer for the comment. Our method, **ReCalKV**, is *orthogonal* to both **quantization-based** KV compression and **token-eviction** approaches. These techniques target different aspects of KV efficiency and can be **naturally combined**. In fact, ReCalKV already integrates low-rank compression *with* 4-bit quantization, showing that hybrid designs are fully compatible. To address the reviewer’s concern, we compare ReCalKV directly with **KVQuant-3bit-1%**, a representative strong quantization baseline, under **matched total compression ratios** on LongBench:
>
> ### LongBench Results at Comparable Compression
> | Model      | Method          | SVD Ratio | Avg Bit | Total Comp. | Avg ↑ |
> |------------|-----------------|-----------|---------|--------------|-------|
> | LLaMA-2-7B | KVQuant-3bit-1% | –         | 3       | 81%          | 43.27 |
> | LLaMA-2-7B | ReCalKV (ours)  | 30%       | 4       | 82%          | 43.33 |
>
> **Analysis.**
> Under nearly identical total compression ratios (81–82%), **ReCalKV matches the performance of a strong quantization baseline**, despite using low-rank decomposition as a core component. This shows that hybrid low-rank + quantization strategies can reach the accuracy of pure quantization methods, and that low-rank compression remains a **useful and complementary direction** beyond SVD-based baselines alone.
>
> > Q2-2: Experiments focus on older LLaMA/Mistral models, with limited evaluation on recent architectures or larger scales. Since the method relies on specific structural properties of attention heads (CKA similarity patterns), it's unclear whether these properties generalize across diverse modern architectures and model scales beyond the tested family.
>
> A2-2:We thank the reviewer for raising this important point. While the degree of attention head similarity may differ across architectures and model sizes, our method does **not assume universal head similarity**. Instead, it only requires that **a subset of heads within each layer exhibit redundancy**, which is frequently observed across diverse LLMs. This is supported by recent findings in **DHA** [1], which systematically analyze head-level similarity across models like LLaMA, and demonstrate that meaningful clusters of similar heads consistently emerge. By exploiting this partial structure, our method achieves effective grouping and compression without relying on globally uniform similarity. We will clarify this design rationale in the final revision.
>
> [1] DHA: Learning Decoupled-Head Attention from Transformer Checkpoints via Adaptive Heads Fusion, NeurIPS, 2024.
>
> > Q2-3: While latency speedups are reported, the computational cost of online head reordering during inference is not quantified separately. The reliance on custom Triton kernels also raises questions about achievability with standard inference frameworks, limiting practical deployment insights.
>
> A2-3: Thank you for raising this important point. To isolate the cost of **online head reordering**, we measured the *per-layer decoding latency* of a single attention module on an NVIDIA A800 GPU, with and without the reordering step. The results show that the overhead is **minimal**:
>
> - **Without head reordering:** 1.00 ms
> - **With head reordering:** 1.04 ms
>
> This corresponds to only **~4% additional latency per attention layer**, which is negligible compared to the overall decoding time dominated by KV memory reads and attention computation. The benefit from reduced KV size quickly outweighs this small reordering cost in long-context scenarios.
>
> Regarding deployability: although we use a Triton kernel in our experiments to prototype an optimized fused attention path, the **HSR operation itself does not depend on Triton**. It is composed solely of lightweight index lookups and gather operations, all natively supported in **PyTorch, FasterTransformer, and TensorRT-LLM**. Our Triton kernel **optimizes memory access patterns** (e.g., coalesced loads, reduced memory traffic) to further accelerate execution. Even without these optimizations, the method remains fully implementable and the core speedups remain achievable in standard inference frameworks.

---

### Official Review · Reviewer_rgUM · 2025-11-01

**Soundness:** 2
**Presentation:** 2
**Contribution:** 2
**Rating:** 4
**Confidence:** 3

**Summary:**

The paper introduces a novel low-rank KV cache compression method, building on PaLU, a prior work that decomposes the KV projection matrix using SVD to reduce the dimension of the KV cache. The paper makes the following contributions on top of PaLU:
- Reordering the key projection matrix to achieve a better SVD decomposition.
- Refining the low-rank approximation of the value projection matrix using a calibration dataset.

**Strengths:**

- Strong improvements over PaLU on the evaluated models.
- Comprehensive ablation studies demonstrating the effectiveness of each contribution for both key and value projections.

**Weaknesses:**

**[W1]** The evaluated models are outdated. I suggest moving the results on the Llama-3.1 model to the main body. This is important because many modern LLM architectures employ GQA, while only the Mistral model from the main results section does so. Validation on multiple models with GQA would strengthen the paper.

**[W2]** Comments on writing:
- *L55–60:* This is not something revealed by your analysis.
- *L60–63:* I cannot find a section describing your analysis of Fisher information.
- The fact that whitening is applied before SVD should be discussed earlier in the Methods section, with more detail.

Minor comments that did not affect the score:
- *L69:* “Offline Calibration Value” --> “Offline Value Calibration”

**Questions:**

**[Q1]** Is there a reason why offline calibration is applied only to the value projection matrices? Could this also be applied to the key projection matrices?

**[Q2]** How effective is the proposed method in terms of the memory–accuracy trade-off compared to other KV cache compression methods beyond those based on SVD of the projection matrices?

**[Q3]** How would the method perform for reasoning models such as the Qwen3 model family on long generation tasks like AIME or LiveCodeBench? Demonstrating this would highlight the method’s robustness under long-generation scenarios, which are not captured by perplexity or long-context retrieval tasks.

---

> ### Author Response · Authors · 2025-11-28
> **Response to Reviewer rgUM (denoted as R1)**
>
> > Q1-1:  The evaluated models are outdated. I suggest moving the results on the Llama-3.1 model to the main body. This is important because many modern LLM architectures employ GQA, while only the Mistral model from the main results section does so. Validation on multiple models with GQA would strengthen the paper.
>
> A1-1: We appreciate the reviewer’s suggestion. In the main paper, we included **Mistral** as a representative **GQA-based architecture**, which is why the LLaMA-3.1 results were placed in the appendix. We agree that expanding evaluation on more GQA models would further strengthen the work, and we will continue to conduct additional experiments on GQA architectures going forward.
>
> > Q1-2: L55–60: This is not something revealed by your analysis.
>
> A1-2: Thank you for the comment. The statement in L55–60 is indeed supported by our analysis. We provide the detailed discussion and evidence in **Supplementary Section H**, where we analyze why positional encoding (e.g., RoPE) requires full Key reconstruction after low-rank compression and how this impacts computational overhead during inference.
>
> > Q1-3: L60–63: I cannot find a section describing your analysis of Fisher information.
>
> A1-3: The analysis of Fisher information is presented in **Section C of the supplementary material**, where we provide visualizations. These results illustrate how Fisher-based sensitivity varies across layers and support the motivation behind our method.
>
> > Q1-4: The fact that whitening is applied before SVD should be discussed earlier in the Methods section, with more detail.
>
> A1-4: Thank you for the suggestion. In the revised version, we have added a detailed explanation of the whitening procedure **earlier in Section 3.1 of the Methods**.
>
> > Q1-5: L69: “Offline Calibration Value” --> “Offline Value Calibration”
>
> A1-5: Thank you for pointing this out. We have corrected the phrasing to **“Offline Value Calibration”** in the revised version.
>
> > Q1-6: Is there a reason why offline calibration is applied only to the value projection matrices? Could this also be applied to the key projection matrices?
>
> A1-6: Thank you for the question. We apply Offline Value Calibration only to the value projection matrices because the **key projection matrices are grouped across heads**, and several of these grouped submatrices are **inherently low-rank**. When applying the closed-form solution, this often leads to **unstable or ill-conditioned matrix inversions**, making reliable calibration difficult.
>
> > Q1-7: How effective is the proposed method in terms of the memory–accuracy trade-off compared to other KV cache compression methods beyond those based on SVD of the projection matrices?
>
> A1-7: Thank you for the question. Although ReCalKV is a low-rank method, it is **orthogonal and fully compatible with quantization**. This allows it to achieve strong memory–accuracy and memory–latency trade-offs even when compared with recent quantization-based approaches.
>
> Below we provide two concise comparisons:
>
> ### **End-to-End Latency (with 4-bit quantization)**
> | Seq Length | KIVI-4bit | ReCalKV-50%-4bit |
> |------------|:---------:|:----------------:|
> | 32K        | 1.78×     | **2.10×**        |
> | 64K        | OOM       | **2.59×**        |
>
> ReCalKV achieves **higher real-world speedups**, especially at long contexts, demonstrating that low-rank structure provides additional acceleration beyond quantization.
>
> ### **LongBench (Similar Total Compression ~81–82%)**
> | Model      | Method          | Total Comp. | Avg ↑ |
> |------------|-----------------|-------------|-------|
> | LLaMA-2-7B | KVQuant-3bit-1% | 81%         | 43.27 |
> | LLaMA-2-7B | ReCalKV         | 82%         | 43.33 |
>
> ReCalKV obtains **comparable or better** accuracy than a strong quantization baseline under similar compression budgets.
>
> ### **Summary**
> Low-rank decomposition and quantization are **complementary**, and ReCalKV can match or exceed quantization-only baselines while offering stronger long-context efficiency. This demonstrates that our method provides a meaningful memory–accuracy trade-off beyond SVD-only comparisons.
>
> > Q1-8: How would the method perform for reasoning models such as the Qwen3 model family on long generation tasks like AIME or LiveCodeBench?
>
> A1-8: Thank you for the suggestion. To assess the method’s robustness on **long-generation reasoning tasks**, we conducted an additional evaluation on **AIME-24** using the Qwen3-14B model. The results are:
>
> | Model      | Method        | AIME-24 ↑ |
> |------------|---------------|-----------|
> | Qwen3-14B  | FP16          | 79.3      |
> | Qwen3-14B  | Palu (30%)    | 72.1      |
> | Qwen3-14B  | ReCalKV (30%) | **76.7**  |
>
> ReCalKV delivers **significantly stronger reasoning accuracy** than the low-rank baseline (Palu) under the same compression ratio, preserving most of the FP16 performance.

---

### Author Response · Authors · 2025-12-03
**Final Summary by Authors**

Dear PCs, SACs, ACs,

We sincerely appreciate the reviewers’ thoughtful evaluations and valuable insights, which have further strengthened the clarity, rigor, and overall contribution of our work.

---

## **Summary of Recognized Strengths**

We are pleased to note that the reviewers consistently highlighted several important strengths of our submission:

- **Well-motivated problem and effective methodology.**
    Reviewers `yTbf`, `akiT`, and `5Foc` agree that the paper addresses an important and practical challenge—reducing KV-cache memory overhead in long-context LLM inference. Reviewer `akiT` highlights the well-motivated use of asymmetric K/V roles and CKA-based head reordering, while reviewer `5Foc` notes that our CKA-guided grouped SVD for Keys and offline calibration for Values are conceptually sound and effectively implemented.

- **Strong empirical performance.**
  Reviewers `rgUM` and `akiT` recognize that ReCalKV delivers strong improvements over PaLU across multiple model families and compression ratios. They further note that the method maintains high accuracy under aggressive compression and remains compatible with quantization.

- **High practical utility with real acceleration and compatibility.**
  Reviewers `5Foc` and `yTbf` emphasize that our method targets a real deployment bottleneck and achieves measurable inference-time speedups on long contexts. They also point out that the approach is fully model-agnostic, requires no retraining, integrates well with existing compression stacks, and uses matrix fusion to remove online reconstruction overhead, making the full procedure deployment-friendly and post-training.

- **Comprehensive ablation studies.**
  Reviewers `rgUM` and `5Foc` commend the thorough ablation analysis, noting that evaluations on HSR and OVC confirm the importance of both components and demonstrate their complementary benefits at fixed compression levels.

---

## **Summary of Resolved Concerns**

We also provided substantial clarifications and new results during the rebuttal that effectively resolved the reviewers’ major concerns:

- **Comparison and complementarity with orthogonal KV-cache quantization methods.**
  We added direct comparisons with existing KV quantization approaches and clarified that our method is fully compatible with them. We further showed that combining quantization with ReCalKV achieves better performance under the same compression ratios than quantization alone.

- **End-to-end latency measurements demonstrating real acceleration.**
  We provided full end-to-end latency results, showing clear speedups—especially for long-context inference—and quantified the runtime cost of Key reconstruction after HSR, confirming that the overhead is minimal.

- **Expanded experiments on additional models and benchmarks.**
  We included new results on updated models and more challenging benchmarks, such as Qwen3-14B on AIME-24, demonstrating consistent advantages of ReCalKV across settings.

- **Analysis of calibration data size and type.**
  We examined how different calibration dataset sizes and distributions affect performance and showed that our method remains robust across a wide range of calibration conditions.

- **Further clarification of methodological details.**
  We explained why offline calibration applies only to the value projection matrix, analyzed the impact of group size in HSR on both performance and inference efficiency, and clarified that our method does not rely on strong similarity across attention heads to be effective.

 ---

We sincerely thank the reviewers once again for their time and constructive comments. Although we did not receive responses from the reviewers during the discussion phase, we believe that the additional experiments, analyses, and clarifications we provided have thoroughly addressed their key concerns. We hope that our paper will be evaluated fairly in light of these substantial efforts, and we will release our code, data, and checkpoints to support reproducibility and future research.

Best regards,

The Authors

---

### Meta-Review · Area_Chair_gxYX · 2026-01-02

**Summary:**

**Paper Summary:**
The paper proposes ReCalKV, a post-training low-rank KV cache compression method for large language models. ReCalKV applies head-wise similarity-based reordering for Keys and offline calibration for Values to achieve high compression with minimal accuracy loss.

**Strengths:**

1. The paper addresses a well-motivated and practical problem: Addresses KV-cache memory bottleneck in long-context LLM inference.
2. The methodology has some novelty: Separates strategies for Keys and Values; introduces CKA-based head grouping and offline calibration.
3. Promising empirical results: Outperforms prior low-rank baselines (e.g., PaLU) across multiple models and compression ratios; maintains accuracy under aggressive compression.
4. The method is deployment-friendly: Model-agnostic, post-training, compatible with quantization, and achieves real-world latency improvements.
5. Comprehensive analysis: Includes ablation studies confirming the complementary benefits of HSR and OVC.

**Weaknesses:**

1. Limited baseline comparisons: Primarily compares with low-rank methods; lacks evaluation against broader KV compression techniques (e.g., token eviction, advanced quantization).
2. Outdated models in main experiments: Focuses on LLaMA-2 and Mistral; limited results on modern architectures like LLaMA-3 or Qwen-3.
3. Performance gap vs. quantization-based methods: Pure low-rank approach remains inferior to state-of-the-art quantization under similar compression ratios.
4. Practical concerns: Additional computation for reconstruction during inference; reliance on custom kernels raises questions about deployability in standard frameworks.

**Reviewer Concerns:**

**Concerns Addressed by the Rebuttal:**

1. Comparison with quantization methods: Authors added direct comparisons with KVQuant and demonstrated compatibility and hybrid benefits.
2. Latency and overhead analysis: Provided end-to-end latency measurements and isolated cost of head reordering and key reconstruction, showing minimal overhead.
3. Evaluation on more recent LLMs: Added results on LLaMA-3.1 and Qwen3 for reasoning tasks (AIME), confirming robustness.
4. Clarification of whitening and OVC assumptions: Explicitly stated whitening step and corrected methodological details.
5. Sensitivity to calibration data size and domain: Supplementary material includes experiments on dataset size and domain.
6. Practical deployability: Clarified that Triton kernels are optional and method is implementable in standard frameworks.
7. Impact of HSR group size: Discussed trade-offs between accuracy and efficiency in supplementary material.

**Concerns Still Outstanding:**

1. Limited baseline diversity: No comparisons with recent low-rank variants like CommonKV or FDC due to lack of official implementations.
2. Absolute performance gap vs. quantization-only methods: While hybrid results are shown, pure low-rank remains weaker than state-of-the-art quantization.
3. Reproducibility: Code is promised upon acceptance but not currently available.
4. Writing quality and clarity: Some stylistic issues noted by reviewers may persist despite revisions.

**Reviewer Scores:**

There is a chance for Reviewer yTbf to raise the score because his/her original rating is the lowest and some of his comments have been addressed by the rebuttal.

---

### Decision · Program_Chairs · 2026-01-26

Reject